# Analyzing Data-Centric Properties for Graph Contrastive Learning

**Puja Trivedi**
University of Michigan
pujat@umich.edu

**Ekdeep Singh Lubana**
University of Michigan
CBS, Harvard University
eslubana@umich.edu

**Mark Heimann**
Lawrence Livermore National Labs
heimann2@llnl.gov

**Danai Koutra**
Unversity of Michigan
dkoutra@umich.edu

**Jayaraman J. Thiagarajan**
Lawrence Livermore National Labs
jjayaram@llnl.gov

## Abstract

Recent analyses of self-supervised learning (SSL) find the following data-centric properties to be critical for learning good representations: *invariance* to task-irrelevant semantics, *separability* of classes in some latent space, and *recoverability* of labels from augmented samples. However, given their discrete, non-Euclidean nature, graph datasets and graph SSL methods are unlikely to satisfy these properties. This raises the question: how do graph SSL methods, such as contrastive learning (CL), work well? To systematically probe this question, we perform a generalization analysis for CL when using generic graph augmentations (GGAs), with a focus on data-centric properties. Our analysis yields formal insights into the limitations of GGAs and the necessity of task-relevant augmentations. As we empirically show, GGAs do not induce task-relevant invariances on common benchmark datasets, leading to only marginal gains over naive, untrained baselines. Our theory motivates a synthetic data generation process that enables control over task-relevant information and boasts pre-defined optimal augmentations. This flexible benchmark helps us identify yet unrecognized limitations in advanced augmentation techniques (e.g., automated methods). Overall, our work rigorously contextualizes, both empirically and theoretically, the effects of data-centric properties on augmentation strategies and learning paradigms for graph SSL.

## 1 Introduction

Self-supervised learning (SSL) [1–9] has revolutionized visual representation learning by producing representations that are more robust [10, 11], transferable [12, 13], and semantically consistent [6] than their supervised counterparts. This impressive empirical success has motivated a surge of efforts that seek to gain insights into SSL's behavior [14–21] or adapt successful frameworks to different modalities, including graph data [22–26]. Notably, many analyses of SSL have converged upon the following data-centric properties as critical to its success: (i) augmentations should induce *invariance* to task-irrelevant attributes, as to better reflect the underlying data generation process and improve generalizability; (ii) samples (and corresponding augmentations) from different underlying classes should be *separable* in some latent space, as to ensure a high-performing classifier is realizable; and (iii) labels of augmented samples should be *recoverable* from the natural sample using which they were generated [16, 20, 27] so that representations are semantically consistent for downstream tasks.

---

Correspondence to pujat@umich.edu.

36th Conference on Neural Information Processing Systems (NeurIPS 2022).

Due to the continuous representation of natural images and well-designed augmentation strategies, these properties are indeed aligned with standard visual SSL practices [28].

However, despite the growing popularity of SSL for graph representation learning, it appears unlikely that the above properties are supported for non-Euclidean, discrete data. Indeed, the design of *recoverable* graph data augmentation [29–31] remains an open research area because is it difficult to determine *prima facie* what changes to a graph's topology or node features will preserve semantics. Moreover, as graphs are often abstract representations of structured data, it is also unclear what *invariances* are relevant to the downstream task. The assumption of a *separable* latent space may also be violated as intermediate points in this latent space may be meaningless in the discrete, structured input space. In contrast to natural image data, the systematic evaluation of these properties for graph SSL is difficult as it must accommodate both discrete and structured data.

*Our Work.* Better understanding the relationship between graph SSL practices and the aforementioned properties can help explain the behavior of existing frameworks and inform the design of new ones. Therefore, in this work, we take the first step by analyzing commonly used generic graph augmentations (GGAs) and designing useful tools that enable probing of these properties, including a theoretical framework and a synthetic data generation process that helps disentangle the effects of unrecoverable augmentations from performance. Our contributions can be summarized as follows:

**Sec. 3: Analysis of Generalization and Separability.** We provide the first generalization error bound for graph CL when using GGAs, demonstrating that GGAs can induce a performance-separability trade-off that is determined by underlying dataset properties (see Figure 1).

**Sec. 4.1: Missing Invariance on Benchmark Datasets.** On standard benchmarks, we show that models trained with GGAs have marginal improvements in accuracy and induce limited task-relevant invariance, at best, when compared to untrained encoders. We thus reveal a fundamental misalignment between the objectives and practical behavior of graph CL (see Figure 2).

**Sec. 4.2: Synthetic Data Generation Process.** We propose a synthetic data generation process that allows for control over augmentation recoverability and dataset separability (see Figure 3). Using this process, we validate our theoretical observations and demonstrate that recently proposed automated and implicit augmentation methods struggle to induce task-relevant invariances (see Figure 4).

## 2  Background

In this section, we briefly discuss existing graph SSL paradigms. (Please see App. G for additional discussion.) We then discuss the motivation behind data-centric properties (task-relevant invariance, separability and recoverabilty) central to this work.

**Self-Supervised Graph Representation Learning.** Recent advancements in representation learning have been driven by the SSL paradigm, where the goal is to ensure representations have high similarity between positive views of a sample and high dissimilarity between negative views. Existing SSL frameworks can be broadly categorized based on the mechanism adopted for enforcing this consistency: contrastive learning (CL) frameworks [1, 8, 7, 22, 29, 31, 32], such as GraphCL[22], use the InfoNCE loss; approaches that rely only on positive pairs, such as SimSiam [2] and BGRL [24] use Siamese architectures with stop gradient [2] and asymmetric branches [21] respectively; SpecCL [15] uses a spectral clustering loss (SpecLoss) to enforce consistency; others attempt to directly reduce redundancy between views [3, 33]. Despite these differences, all methods rely upon data augmentation to generate positive views, which are assumed to share semantics. Generic graph augmentations (GGAs) [22] are a popular strategy and assume limited changes to a graph's node features or topology are unlikely to alter its label. GGAs include random node dropping, edge perturbation, masking node attributes and sampling subgraphs. Other strategies include using diffusion matrices [23], GGAs with a non-uniform prior, automated methods which rely upon bi-level optimization [29] or adversarial optimization [31], and implicit methods, such as SimGRACE [32], which use weight-space perturbations as augmentations. Here, we primarily focus on GGAs due to their popularity, simplicity and effectiveness. Please see App. G for additional discussion about augmentation paradigms.

**Theoretical Analsyis of SSL.** Several different perspectives have recently been used to successfully analyze SSL's behavior, including learning theory [15, 14, 34], causality [18, 17], information theory [27], and loss landscapes [35–38]. Many of these analyses assume, either implicitly [18, 34] or

explicitly [15, 28, 39, 40], the existence of a latent space that is *invariant* to augmentation functions and supports the properties of *recoverability* and *separability* (also see Figure 1).

*Invariance to Augmentations:* Producing similar representations for positive views, i.e., augmentations, induces invariance to the corresponding transformation function. Indeed, if augmentations are related by properties that are *not relevant* to the downstream task, representations will become *invariant* to this relationship over the course of SSL training and generalization will improve [41, 16]. Conversely, however, if augmentations induce invariance to *relevant* properties, then representations will fail to represent this information and are likely to lose task performance (e.g, color invariance is harmful when classifying different Labradors) [20, 42]. This latter point is often ignored by the theoretical analyses mentioned above. We note Tian et al.'s information theoretic framework [16] is a notable exception to this critique; we discuss the limitations of their results in App. C.3.

*Recoverability and Separability:* These properties state that in the latent space which instantiates the data generation process, two augmentations of a sample are close to each other (e.g., a clear and blurry dog) and unrelated points (e.g., dogs and cats) are sufficiently separated from each other. It is often implicitly assumed that only task-relevant augmentations are allowed [15, 28]. While originally proposed for manifolds [39], both recoverability and separability have been recently converted to graph connectivity properties [15] and verified empirically on modern deep learning methods [28]. Specifically, recoverability and separability can be used to bound generalization error on unseen data and we demonstrate how this can be done for graph CL in Sec. 3.

**Notations.** Let $\overline{\mathcal{X}}$ be a natural dataset with $r$ different classes. Our use of word natural implies the samples in this dataset were collected via a natural sensing process (e.g., molecules from wet-lab experiments or scene graphs from images). We use $\mathcal{A}(.|\overline{g})$ to denote the distribution of augmentations for the sample $\overline{g} \in \overline{\mathcal{X}}$. Here, $\mathcal{A}(g|\overline{g})$ represents the probability of generating a particular augmentation $g$, and $\mathcal{X} := \cup_{\overline{x} \in \mathcal{P}_{\overline{\mathcal{X}}}} \mathcal{A}(\cdot|\overline{g})$ is the set of all samples transformed via our set of augmentation functions. Let $f : \mathcal{X} \rightarrow \mathbb{R}^d$ be a feature extractor, where $f(x)$ can be used for downstream tasks. Unless otherwise noted, let $\overline{g}$ be a natural (graph) sample from $\overline{\mathcal{X}}$, $\mathcal{A}(\cdot|\cdot)$ be some GGA, and $g \sim \mathcal{A}(\cdot|\overline{g})$ be an augmented graph generated using a given GGA. $\mathcal{V}_{\overline{g}}$ and $\mathcal{E}_{\overline{g}}$ correspond, respectively, to the node and edge sets of $\overline{g}$. We note our generalization analysis will specifically focus on the recently proposed contrastive loss by HaoChen et al. [15], called SpecLoss ($\mathcal{L}(f)$), which we define as follows: let $g \sim \mathcal{A}(\cdot|\overline{g})$, $g^+ \sim \mathcal{A}(\cdot|\overline{g})$, given $\overline{g} \in \overline{\mathcal{X}}$, and $g^- \sim \mathcal{A}(\cdot|\overline{g}')$, given $\overline{g}' \sim \mathcal{P}_{\overline{\mathcal{X}}} \wedge \overline{g}' \neq \overline{g}$. Then, for the positive/negative pairs $(\overline{g}, g^+)/(\overline{g}, g^-)$, SpecLoss is: $\mathcal{L}(f) = -2 \cdot \mathbb{E}_{\overline{g}, g^+} \left[ f(\overline{g})^\top f(g^+) \right] + \mathbb{E}_{\overline{g}, g^-} \left[ \left( f(\overline{g})^\top f(g^-) \right)^2 \right]$. In a contemporary work, Saunshi et al. [14, 41] developed a generalization analysis for general contrastive loss functionals, including SpecLoss. Our analysis has a similar algorithmic flow as Saunshi et al.'s and hence the takeaways from our work can be easily extended for other contrastive methods as well. We provide additional discussion of this extension in App. C.1.

## 3 Generalization Bounds for CL with GGA

As discussed above, recent analyses have found that SSL generalization error can be bounded under the assumptions of invariance to relevant augmentations, recoverability, and separability. In this section, we demonstrate how GGAs influence these properties by deriving a generalization bound tailored for graph data. Notably, this bound allows us to demonstrate conditions where using GGAs results in low separability and recoverability, motivating the need for augmentations that induce task-relevant invariances that go beyond generic perturbative graph transformations.

**Key Insight:** Our main idea for the following analysis is that *GGAs can be instantiated in a general manner as a composition of graph edit operations*. This allows us to derive a unifying assumption related to recoverability and separability in terms of the graph edit distance (GED) between samples. Moreover, because GED amongst samples is a property intrinsic to the dataset, we can now discuss how the tightness of a SSL generalization error bound (SpecLoss's, specifically) will change as a function of GED between samples of underlying classes and augmentation strength.

We begin by defining GED and explaining how GGAs can be represented using graph edit operators.

**Definition 3.1** (Graph Edit Distance)**.** Let the elementary graph operators comprise the set of graph edits: these include *node insertion*, *node deletion*, *edge deletion*, *edge addition*, and an additional

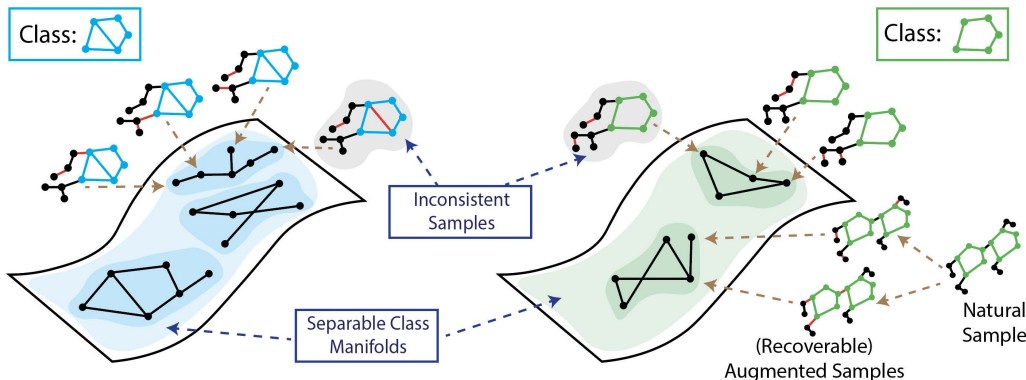

Figure 1: **Illustrating data-centric properties forming the core of our assumptions.** Our generalization analysis (Sec. 3) relies upon several data-centric properties, namely recoverability, separability, and frequency of inconsistent samples. Here, we illustrate these properties via a figure. (i) **Separability:** Samples from different classes should be *separable*, as illustrated by the existence of separate manifolds for different classes. This property helps assume the existence of a classifier $h$ that can classify natural samples with low error. (ii) **Recoverability:** Labels of augmented samples should be *recoverable* from the original samples from which they were generated. This entails that augmentations generated from the same original samples are expected to be closer in latent space than two arbitrary samples, which will likely correspond to different classes. This property helps assume a constraint on the classifier $h$ that it must also classify the augmentations of a sample to the same class as that of the sample. (iii) **Inconsistent Samples:** While the likelihood of generating augmentations that alter class semantics is low for image data, this if often note the case in graphs, especially when using generic graph augmentations. We refer to augmentations that can be generated from original samples belonging to different classes as *inconsistent*, and demonstrate that graph edit distance can be used to identify such samples. Overall, our theory shows inconsistent samples decrease separability and recoverability, harming generalization. (Figure inspired from Chung et al. [43] and HaoChen et al. [15].)

*categorical feature replacement* operator. Then, $GED\,(g_1, g_2) = \min_{(e_1,\ldots,e_k) \in \mathcal{P}(g_1,g_2)} \sum_{i=1}^{k} c\,(e_i)$, where $\mathcal{P}\,(g_1, g_2)$ is the set of paths (series of edit operations) that transforms graph $g_1$ to be isomorphic to graph $g_2$. Here, $e_i$ is $i$-th edit operation in the path, and $c(e_i)$ is the cost for performing the edit.

As shown in Table 1, frequently used GGA transforms can be easily decomposed using standard graph edit operators described in Def. 3.1. For example, assuming each operator has a unit cost, the edge perturbation augmentation can be seen as applying the minimum cost path consisting of edge deletion and edge addition operations to obtain $g$ from $\overline{g}$. Further, augmentation strength and the set of possible augmentations for a given natural sample can also be expressed in terms of GED:

**Lemma 3.2.** *Allowable augmentations can be expressed using GED. Let $\gamma$ represent augmentation strength or the fraction of the graph that GGAs may modify. Then, $\delta \in \{\lfloor \gamma |\mathcal{V}_{\overline{g}}| \rfloor, \lfloor \gamma |\mathcal{E}_{\overline{g}}| \rfloor\}$ represents the number of discrete, allowable modifications for the specified GGA, so $GED(\overline{g}, g) \leq \delta$. Correspondingly, we have $g \in \mathcal{A}(\overline{g}) \Leftrightarrow GED(g, \overline{g}) \leq \delta$.*

For example, consider a graph $g \sim \mathcal{A}(\cdot|\overline{g})$, generated via node dropping. Then, $g$ contains $1 - \delta$ nodes and the minimum cost path to transform $\overline{g}$ to $g$ contains only $\delta$ "node deletion" operations. Further, all augmentations generated from $\overline{g}$ will have $1 - \delta$ nodes and thus have $GED(\overline{g}, g) \leq \delta$. In Appendix D, we prove the above statement and demonstrate how to approximate $|\mathcal{A}(\overline{g})|$ (e.g., the set of allowable augmentations for a given natural sample) using a combinatorial, counting

Table 1: **Generic Graph Augmentations vs. Graph Edit Operators.** GGAs can be straightforwardly expressed using graph edit operators. Please see Appendix D for a detailed discussion.

| Augmentations | Graph Edit Operators |
|---|---|
| Node Dropping | Node Deletion |
| Edge Perturbation | Edge Deletion, Edge Addition |
| Categorical Attribute Masking | Feature Masking Operator |
| Sub-graph Sampling | Node Deletions |

procedure that is dependent on $\delta$. Because GGAs are applied randomly, note that the probability of a generating a particular augmentation is $\mathcal{A}(g|\overline{g}) \approx \frac{1}{|\mathcal{A}(\overline{g})|}$. Given these definitions, we now derive a unifying assumption in terms of GED between samples. We begin by formally introducing the separability and recoverability assumptions, focusing on the recently proposed, unified version [15]:

**Assumption 3.3** (Separability plus Recoverability Assumption, (Assm. 3.5 in [15])). Let $\overline{g} \in \overline{\mathcal{X}}$ and $y(\overline{g})$ be its label, and $g \sim \mathcal{A}(\cdot|\overline{g})$. Assume that there exists a classifier $h$, such that $h(g) = y(\overline{g})$ with probability at least $1 - \alpha$. We refer to $\alpha$ as the error of $h$.

See Figure 1 a visualization explaining this assumption. Intuitively, Assm. 3.3 states that there must exist a classifier $h$ that is able to associate a sample's label with its augmentations, hence enabling recoverability, i.e., representations of augmentations are close to each other. Meanwhile, by ensuring augmentations of samples from a class with label "A" are classified as "A" and from a class with label "B" are classified as "B", the assumption simultaneously enables separability, i.e., representations of samples from different classes should be dissimilar. As we will see, the generalization bound will be a function of $\alpha$, the probability that a classifier satisfying Assm. 3.3 associates augmentations of a class's samples with another class. As $\alpha$ grows larger, the generalization error bound becomes less tight. Therefore, it is important to understand how the choice of augmentation and augmentation strength ($\gamma$) can influence the error of $h$. We show one can also understand $\alpha$ as a trade-off between inter-class GED of samples and augmentation strength.

Intuitively, $h$ will incur error on augmented samples that can be generated from a set of natural samples that belong to different underlying classes, as it is unclear how these samples should be embedded in a latent space. We now formally define such samples. First, using Lemma 3.2, we can determine if two augmentations could have been generated from the same sample.

**Corollary 3.4.** *(Co-occuring augmentations.) Let $\overline{g} \in \overline{\mathcal{X}}$ and $g, g' \in \mathcal{X}$. Then, $g \sim \mathcal{A}(\overline{g}) \wedge g' \sim \mathcal{A}(\overline{g}) \Leftrightarrow GED(g, g') \leq 2\delta$, where $\delta = \min\{\lfloor \gamma|\mathcal{V}_{\overline{g}}|\rfloor, \lfloor \gamma|\mathcal{E}_{\overline{g}}|\rfloor \lfloor \gamma|\mathcal{V}_g|\rfloor, \lfloor \gamma|\mathcal{E}_g|\rfloor\}$.*

Given the above result, we now define inconsistent samples as follows.

**Definition 3.5** (Inconsistent Samples). Let $g \in \mathcal{X}$, and $y : \overline{\mathcal{X}} \to r$ be a labeling function. Further, let $\overline{\mathcal{X}}_{in} = \{\overline{g}|\overline{g} \in \overline{\mathcal{X}} \wedge GED(g, \overline{g}) \leq \delta\}$ be the set of natural samples that may have generated $g$ and $Y^*_{in} = \{y(\overline{g})|\overline{g} \in \overline{\mathcal{X}}_{in}\}$ be the set of unique labels. If $g$ is an inconsistent sample, $|Y^*_{in}| > 1$.

Essentially, if two augmentations co-occur (see Corr. 3.4) from two or more *different* natural samples, such that the samples *do not* share the same underlying label, we refer to such samples as *inconsistent* (also see Figure 1). Now, we assume the behavior of $h$ on inconsistent samples is fixed such that $h(g) = y$, for some fixed $y \in Y^*_{in}$. This allows us to use $h$ to induce a $r$-way partition over $\mathcal{X}$, such that each sample, $g$, belongs to a partition, $\mathbf{S}_h(g)$. Further, because $h$ incurs error on inconsistent samples, $\alpha$ can be lower bounded by the ratio of inconsistent to total samples. To this end, we use GED to identify inconsistent samples by identifying disagreement amongst partitions as follows.

**Lemma 3.6** (Using GED to identify inconsistent samples). *Let $g, g' \in \mathcal{X}$ and $GED(g, g') \leq 2\delta$ such that $g \in \mathbf{S}_i \wedge g' \in \mathbf{S}_j$ and $i \neq j$, where partitions are induced by $h$. Then, at least one $\tilde{g} \in \{g, g'\}$ must be an inconsistent sample.*

Note that the above lemma does not rely on ground-truth label information to identify inconsistent samples, but only GED from natural samples. Given that the error on inconsistent samples is irreducible, as it is unclear which $y \in Y_{in}$ is correct, we can lower bound the error of $h$ as follows:

**Corollary 3.7** (Error bound due to Inconsistent Samples). *The error of $h$ can be lower-bounded as*

$$\alpha \geq \frac{\sum_i^r \sum_{g \in S_i, g' \notin S_i} \mathbb{1}(GED(g, g') \leq 2\delta)}{|\mathcal{X}|}. \tag{1}$$

Here, the number of inconsistent samples can be approximated via $\sum_i^r \sum_{g \in S_i, g' \notin S_i} \mathbb{1}(GED(g, g') \leq 2\delta)$ and $|\mathcal{X}|$ can be estimated using a combinatorial counting procedure. Thus, the above corollary reflects the fact that error on inconsistent samples cannot be reduced due to label un-identifiability.

As mentioned before, the generalization bound by HaoChen et al. [15] for SpecLoss is a function of $\alpha$. Deriving a lower bound on $\alpha$ will allow us to comment exactly when error is likely to become vacuous. To this end, we need a final definition of *partition dissimilarity* that induces a notion of clustering of similar datapoints in our analysis.

**Definition 3.8** (Partition Dissimilarity). Let $S_1, \ldots, S_r$ be an $r$-way partition of $\mathcal{X}$. Then, we define the partition dissimilarity for a given partition as

$$\phi_{\mathcal{X}}(S_i) = \frac{\sum_{g \in S, g' \notin S} \mathbb{1}(GED(g, g') \leq 2\delta)}{\sum_{g \in S} |\{g'|GED(g, g') \leq 2\delta\}|}. \tag{2}$$

Intuitively, we use the partitions induced by $h$ as a proxy for class labels and co-occurrence as a notion of similarity (see Lemma 3.2). Then, the quality of the partition is determined by the ratio

of the samples that belong to a given partition, but are also similar to samples from other partitions, to the total number of samples that are close to the partition. Note that partition dissimilarity is an often studied term in clustering problem and a general version of conductance, the property used for spectral clustering on a similarity graph which forms the basis of SpecLoss [15].

We are now ready to state our main result that re-derives the generalization error of SpecLoss in terms of GGAs, using the definitions of co-occurring pairs (Def. 3.4) and dissimilar partitions (Def. 3.8). Notably, we will decompose bound in terms of the number of co-occurring augmentation-pairs within the same partition and the number of pairs that cross partitions, which are defined respectively as, $\lambda = \sum_{\boldsymbol{g} \in S_*, \boldsymbol{g}' \in S_*} \mathbb{1}(GED(\boldsymbol{g}, \boldsymbol{g}') \leq 2\delta)$, and $\mu = \sum_{\boldsymbol{g} \in S_*, \boldsymbol{g}' \notin S_*} \mathbb{1}(GED(\boldsymbol{g}, \boldsymbol{g}') \leq 2\delta)$.

**Theorem 3.9** (Generalization Bound for SpecLoss with GGA). *Assume the representation dimension $k \geq 2r$ and Assm. 3.7 holds for $\alpha \geq 0$. Let $F$ be a hypothesis class containing a minimizer $f_{pop}^*$ of SpecLoss, $\mathcal{L}(f)$, which produces a $\lfloor k/2 \rfloor$-way partition of $\mathcal{X}$ denoted by $\{S_*\}$. Let its most dissimilar partition have dissimilarity denoted by $\rho_{\lfloor k/2 \rfloor} = \min_i \phi(S_i \in \{S_*\})$. Then, $f_{pop}^*$ has a generalization error bounded as:*

$$\mathcal{E}(f_{pop}^*) \leq \widetilde{O}\left(\alpha/\rho_{\lfloor k/2 \rfloor}^2\right) = \widetilde{O}\left(\frac{r}{|\mathcal{X}|}\left[\mu + 2\lambda + \frac{\lambda^2}{\mu}\right]\right), \tag{3}$$

**Discussion.** By deriving expressions for $\alpha$ and $\phi$ as well as equivalently representing the original bound in terms of the more intuitive expressions, $\mu$ and $\lambda$, we can gain insights into several empirical and intuitive observations in graph CL. We will study these points further in Sec. 4.2 via a synthetic dataset that was motivated from the analysis above and allows more fine-grained evaluation.

*Invariance and Relevance of Augmentations.* GGAs assume that limited changes to a graph's structure will not alter its semantics and aggressively increasing augmentation strength will eventually harm generalization. However, through our analysis, we see that the generalization error bound is non-decreasing with respect to $\delta$ when $\frac{\lambda^2}{\mu} \leq \mu$, i.e., the number of cross partition pairs dominates the expression, as this ratio depends on $\delta$. Indeed, for some $\delta' = \delta + \epsilon$, where $\epsilon > 0$, $\mu_{\delta'} = \sum_{\boldsymbol{g} \in S_i, \boldsymbol{g}' \notin S_i} \mathbb{1}(GED(\boldsymbol{g}, \boldsymbol{g}') \leq 2\delta) + \sum_{\boldsymbol{g} \in S_i, \boldsymbol{g}' \notin S_i} \mathbb{1}(2\delta \leq GED(\boldsymbol{g}, \boldsymbol{g}') \leq 2\delta + \epsilon) = \mu_\delta + \sum_{\boldsymbol{g} \in S_i, \boldsymbol{g}' \notin S_i} \mathbb{1}(2\delta \leq GED(\boldsymbol{g}, \boldsymbol{g}')) \leq 2\delta + \epsilon$. Thus, the number of cross partitions is always non-decreasing with respect to $\delta$. Thus, *we clearly see that when augmentations are agnostic of the task, their corresponding invariances yield poor representations with vacuous generalization.*

*Separability.* Our analysis also demonstrates that the success of a particular augmentation strength is dependent on the GED between samples belonging to different classes. Given that inter-class GED is an intrinsic dataset property that proxies dataset separability, this implies that there are combinations of datasets and augmentation strengths for which GGAs will necessarily incur vacuous bounds, even for low augmentation strengths. In such settings, augmentations that improve recoverability and induce task-relevant invariances are necessary to improve downstream task performance. While many works have conjectured that task-relevant graph augmentations will improve performance, ours is the first to demonstrate why they are needed. Indeed, in Sec. 4.1, we find that GGAs are unable to induce such invariances on benchmark datasets.

*Recoverability.* As shown in Thm. 3.9, better recoverability will improve the tightness of the generalization bound. However, we see that from Coll. 3.7, that recoverability will only decrease as $\delta$ increases and as discussed above, there exist datasets where GGAs are not amenable. This further motivates the need for task-relevant augmentations so that the effects of poor augmentations are disentangled from method performance.

## 4 Experimental Verification

In this section, we conduct experiments using both standard benchmarks (Sec. 4.1) and our proposed synthetic dataset generation process (Sec. 4.2) to empirically validate our theoretical conclusions.

### 4.1 A Closer Look at the Effectiveness of Invariance to GGA

In Sec. 3, we demonstrated GGAs can harm generalization by influencing recoverability and separability. Though computing these properties directly is intractable on benchmark datasets, our analysis for graph datasets and prior works on vision [17, 18, 20] show that if augmentations induce invariances that are *task-relevant*, downstream error should reduce. This corresponds to meaningfully

related samples having similar representations (recoverable) and unrelated samples having dissimilar representations (separable). However, by using augmentations that perturb topology or features constrained to a small fraction of the original graph, existing graph SSL methods assume such perturbations are relevant to the downstream task. If this is the case, our analysis suggests we should see improvement in performance with increased invariance; else, we will witness no tangible correlation.

*Experimental Setup:* We evaluate seven graph SSL methods on seven, popular benchmark datasets. Specifically, we use the following representative algorithms: (i) *GraphCL* [22], a popular and effective graph CL method; (ii) *GAE*, Graph Autoencoder [44] that uses a reconstruction cost to learn representations; (iii) Augmentation-Augmented Autoencoder [45], which we adapt to graphs to create the Augmentation Augmented Graph Autoencoder (*AAGAE*) that minimizes the reconstruction error between the reconstruction for an augmented sample and the original; (iv) *SpecCL*, which uses the SpecLoss [15] for contrastive training; (v) *SimSiam* [2], a positive-sample-only framework that uses stop gradient; (vi) *BYOL* [21], which avoids negatives samples by using asymmetric branches alongside a stop gradient operation; and (vii) *Untrained representations*, which have been observed to be surprisingly competitive baselines for graph-based learning [46, 47, 31, 42]. To the best of our knowledge, ours is the first work to evaluate AAGAE and SpecCL for graph SSL. We use the same augmentations and encoder architecture as GraphCL. We add a straight-through estimator [48] to GAE/AAGAE's decoder for better training. See Appendix F for further details.

**GGAs fail to induce task-relevant invariance on standard benchmarks.** To measure whether augmentations have induced invariance, we measure recoverability using the representational similarity measures introduced by Wang and Isola [19]. Called Alignment and Uniformity, the two measures are a generalized version of the InfoNCE loss and also encompass other contrastive losses, such as SpecLoss. Formally, alignment is defined as: $\mathcal{L}_{\text{align}}(f; \mathcal{A}) \triangleq \mathbb{E}_{(g,g')\sim\mathcal{A}(\cdot|\bar{g})}\left[\|f(g) - f(g')\|_2^2\right]$. To determine if the invariance is task-relevant, we determine if improved alignment is indicative of improved performance with respect to an untrained baseline model.

*Results.* Fig. 2 shows the difference in invariance and $k$NN with respect to an untrained model's accuracy, averaged over 10 seeds. As can be seen, there is not noticeable correlation between

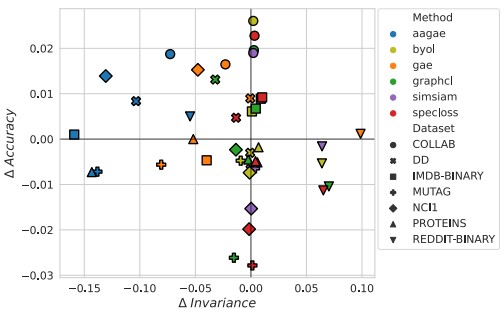

Figure 2: **Invariance vs. KNN Acc.** The change in invariance (Inv.) and accuracy w.r.t. to an untrained model is plotted, where Inv. is measured according to [19]. We see: Inv. has not significantly increased for many datasets/methods, improved Inv. does not necessarily entail better performance (see Reddit), and AAGAE/GAE often sees decreased Inv., likely due to use of a decoder.

invariance and accuracy, especially with respect to the untrained baseline. Notably, on the Reddit dataset, all methods have improved invariance, but do not have significantly better kNN accuracy. Overall, this experiment demonstrates that learning invariance to GGAs is both difficult and often unrelated to task performance, clearly indicating the GGAs struggle to induce task-relevant invariances and do not support recoverable, separable latent spaces needed for good generalization. Moreover, given that GGAs have unknown recoverability on standard datasets, and that trained models were not able to sufficiently outperform untrained baselines, there is need for new datasets where it is possible to go beyond GGA and where we can better understand the merits of different graph SSL paradigms.

## 4.2 Evaluating Graph SSL Methods in a Controlled Setting

Our analysis indicates the role played by recoverability and separability under task-relevant invariances dramatically influences generalization performance. However, given our results that GGAs do not enable these properties and the fact that task-relevance is difficult to define on existing benchmark datasets, empirical verification of our claims requires a dataset that directly enables control over the data generation process. We thus introduce a synthetic dataset that allows us to illustrate how invariance and class separability must be jointly considered when designing augmentations.

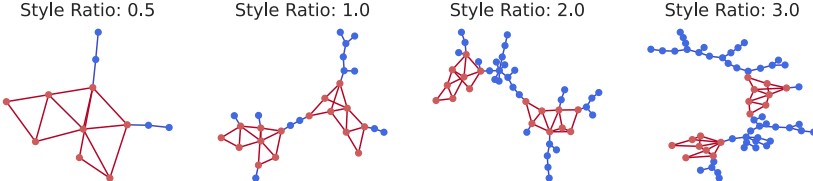

Figure 3: **Synthetic Dataset Generation.** A class-specific motif completely determines the label, and is therefore considered "content". To vary the amount of style, the size of the background tree graph is a ratio of the number of "content" nodes. Our dataset goes beyond binary benchmarks and allows for content-aware augmentations, a critical component to understanding graph SSL.

### 4.2.1 Synthetic Data Generation Process

Given that standard benchmark datasets and augmentation practices are uninformative when evaluating the recoverability and invariance of augmentations, we propose a synthetic data generation process that allows us to understand how the data-dependent assumptions of SSL hold for graph datasets. This process not only enables oracle augmentations where recoverability is known, but also allows us some control over dataset separability.

Our synthetic dataset generation process is designed in accordance to a latent variable model which assumes that the underlying data generation latent representation space can be partitioned into *style* and *content*. Here, *style* represents information that is irrelevant to the downstream task and can be perturbed (i.e., augmented) without changing sample semantics, while *content* represents task-relevant information and should be preserved. We note that while von Kügelgen et al. [17] used the same latent variable model to demonstrate that SSL with data augmentation is able to recover features which disentangle *style* vs. *content*, our objective for using this perspective is to develop a grounded benchmark that provides adjustable knobs over content (task-relevant) and style (task-irrelevant) information. These knobs allow us to understand how data-centric properties affect the performance of different graph SSL algorithms (see Fig. 4). While designing content-aware augmentations for arbitrary graph datasets is a hard problem [42], with oracle knowledge of the data generation process, we can evaluate content-aware augmentations (CAAs) with high recoverability at varying levels of separability, which we approximate through different style levels.

**Generation Process:** The proposed data generation process has three components: a set of $C$ motifs, $\mathcal{M}$, that uniquely determine $C$ classes; a random graph generator, $RBG(n)$, parameterized by the number of nodes (we can equivalently define this based on number of edges); and $\kappa$, the style multiplier, which controls how much irrelevant information a sample contains. To generate a sample, we attach a randomly generated background graph (*i.e.*, style component) to a motif (*i.e.*, content) according to the style multiplier. This simple process addresses several limitations often encountered in graph CL evaluation. Specifically, it (i) allows for varying levels of content-aware augmentation (*i.e.*, edges that can be perturbed in the background graph without harming the motif); (ii) is easily extended beyond binary classification; (iii) contains relatively large number of samples; and (iv) offers a natural test bed for GNN size generalization or transfer learning [49].

### 4.2.2 Difficulties in Recovering Style Invariant Representations

Several real graph datasets can be understood through a style vs. content perspective. For example, in drug discovery tasks [50], molecules can be split into functional groups (content) and carbon rings or scaffold structure (style). One may thus ask: how does varying levels of style vs. content affect the performance of graph URL algorithms, and how do different algorithms benefit from the use of content-aware augmentations? To answer these questions, we conduct the following experiment:

**Experimental Setup.** Let $C = 6$, $\kappa = 4$ and define $RBG(n)$ through a random tree generator, where $n$ is number of the nodes belonging the motif, scaled by $\kappa$. Node features are a constant 10-dimensional vector. To increase task difficulty, we randomly insert between 1-3 motif copies into each sample. Using the specified instaniation of the generation process, we train GraphCL, AAGAE, GAE, and SpecLoss with *content-preserving* edge dropping and random edge dropping at 20% and 60% augmentation strength. We also evaluate two recently proposed automated augmentation methods, JOAO [29] and AD-GCL[31], as well as SimGRACE [32], which uses implicit, weight space perturbations. JOAO is trained with a GGA prior and an expanded GGA prior that includes

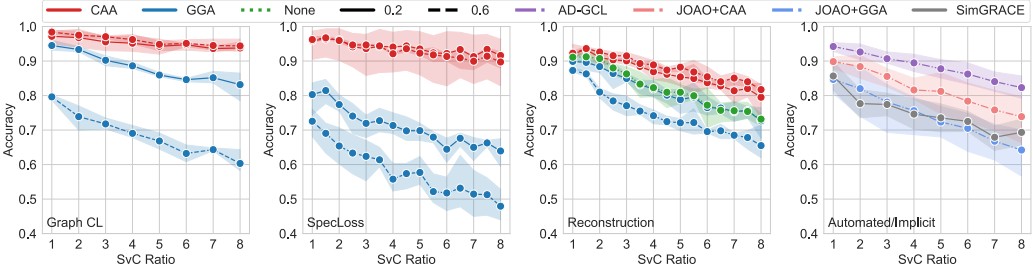

Figure 4: **Style Invariance over Paradigms:** We evaluate several SSL algorithms with different augmentation paradigms and changing style vs. content ratios. We find several notable results: (i) CAAs induce style invariance in contrastive methods, but GGAs do not; (ii) reconstruction methods do not recover task-relevant invariances, even when using CAAs; and (iii) advanced augmentations methods (AD-GCL, JOAO, SimGRACE) lose performance as style increases, indicating they do not induce style-invariance.

content-preserving edge dropping. AD-GCL is trained using a learnable edge-dropping augmentor. A 5-layer GIN encoder is used and models are trained for 60 epochs using Adam (with a learning rate of 0.01). After training, all models are evaluated using the linear probe protocol [1] at varying style ratios. Given that style information is not relevant to the downstream task, we expect models that have truly learned invariance to this information will retain strong performance across different ratios. See Appendix F for more model and training details.

**Results.** We make the following observations using Fig. 4, which clearly demonstrate the value of the proposed benchmark in studying the behavior of different SSL and augmentation paradigms. (i) In accordance to Sec. 3, we empirically see that both GraphCL and SpecLoss do not loss performance as the style ratio increases when using CAAs, indicating the model has learned task-relevant invariances. (ii) Auto-encoding reconstruction methods are an alternative SSL paradigm, but unfortunately also struggle to recover style-invariant solutions. Moreover, the use of the CAAs with such methods does not improve performance as effectively as in contrastive paradigms. (iii) For the first time, we are able to evaluate whether automated methods, which aim to recover strong augmentations without expensive hyper-parameter tuning or hand designing, are able to recover an optimal augmentation that generalizes across style ratios. Unfortunately, we see both AD-GCL [31] and JOAO [29] lose performance as the style ratio increases, indicating such a solution has not been found. Indeed, JOAO is unable to find such a solution even when the augmentation prior includes the oracle CAAs. These results not only highlight the brittleness of such automated methods, but indicate our benchmark is a necessary testbed for such methods. (iv) To avoid corrupting graph semantics when using input-space augmentations, SimGRACE [32] instead uses implicit, weight-space augmentations. However, we find, despite tuning the perturbation parameter, SimGRACE cannot recover strong, style-invariant performance. Overall, using our grounded synthetic benchmark, we are not only able to able to compare the performance of graph SSL algorithms when data-centric properties are supported (e.g., recoverable augmentations), but are also able to identify limitations of advanced augmentation methods that were not apparent using standard benchmarks.

### 4.2.3 Invariance vs. Separability

We now use our synthetic benchmark to investigate how augmentation recoverability influences the balance of invariance and separability in the learned latent space. Considered in isolation, invariance can be trivially satisfied through representation collapse, i.e., all samples are mapped to highly similar representations. However, such representations are not separable as they cannot meaningfully distinguish classes. Therefore, in the following experiment, we jointly consider these properties to understand the benefits of CAAs.

**Experimental Setup.** Using a synthetic dataset at $\kappa = 6$, we respectively train GraphCL with *content-preserving* and random edge dropping at 20% aug-

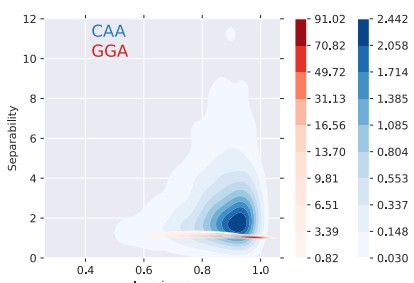

Figure 5: **Invariance vs. Separability**. On our synthetic data with style-to-content ratio $\kappa = 6$ and 20% augmentation strength, GraphCL trained with random augmentations produces representations with high invariance but low separability. In contrast, using content preserving augmentations leads to almost as high invariance, but much greater separability.

mentation strength. We compute an invariance score

for each natural sample by computing the average cosine similarity of its representation with that 30 different augmentations. We compute a separability score by dividing the maximum cosine similarity to a sample of the same class by the maximum cosine similarity to a sample of another class.

**Results.** Figure 5 shows kernel density estimates of the number of samples that have a given invariance and separability, when training with GGA or CAA. GGA induces representations with somewhat higher invariance but much lower separability scores, suggesting some representation collapse are occurred. Indeed, with a higher augmentation strength (60%), we found that using GGA produced invariance and separability scores very close to 1 for all samples, indicating strong collapse. On the other hand, CAA helps GraphCL achieve over an order of magnitude higher separability and still preserves comparably high invariance. We observed similar trends for SpecLoss.

**Invariance vs. Separability in Realistic Settings.** In App. C.2, we replicate this experiment using BACE [51], a molecule-protein interaction dataset, and the biochemistry-based augmentations proposed by Sun et al. [52] as CAAs. We find that our observations continue to hold in this real-world use-case, demonstrating the generality of our theory and practicality of our synthetic benchnmark.

## 5   Conclusion

In this work, we rigorously contextualize, theoretically and empirically, the role of data-dependent properties for graph CL. We propose a novel generalization analysis which, for the first time, formalizes the limitations of using GGAs in graph CL. As we note in Sec. 3, our results can be extended to other contrastive frameworks by leveraging our insight on representing graph augmentations as composable graph-edit operations and extending the contemporary work of Saunshi et al. [41]. We suspect a similar extension can also be made for predictive methods like BYOL by using the analysis of Wei et al. [28] (see App. C.1 for further discussion). In line with our theory, we empirically demonstrate that GGAs fail to induce useful task-relevant invariances on standard benchmarks. We note our empirical results already demonstrate the generality of our results across different methods. Moreover, our insights motivate the design of a principled synthetic benchmark that provides a controlled setting for studying the role of data-dependent properties in graph SSL. Our benchmark also serves as a useful testbed for evaluating the abilities of automated or implicit augmentations techniques. Given the shortcomings we illustrate for such methods on synthetic datasets, we argue the development of domain specific strategies [52] may be a more fruitful direction for future work.

## Acknowledgements

This work was performed under the auspices of the U.S. Department of Energy by the Lawrence Livermore National Laboratory under Contract No. DE-AC52-07NA27344, Lawrence Livermore National Security, LLC.and was supported by the LLNL-LDRD Program under Project No. 21-ERD-012. PT was an intern at Lawrence Livermore National Labs while working on this project. ESL was partly supported via NSF award CNS-2008151.

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
