## A Contributions

PT: Led project formulation, writing, designing & running experiments, and discussion. PT originally conceived of representing generic graph augmentations using composable, graph edit operations to derive a generalization bound based on SpecLoss and made early attempts at this derivation, as well as its interpretation. ESL: Contributed to project formulation, writing, experimental design, and discussion. ESL led theory section, deriving Defn. 3.8 (partition dissimilarity) and Thm 3.9 (generalization bound). ESL and PT refined the analysis together. ESL and PT jointly conceived of using the synthetic dataset and corresponding experiments. PT led the corresponding section. MH: Contributed to running experiments, discussion, writing, and figure generation. DK:assisted in early project ideation. JJT: senior advisor, contributed to project formulation, discussion, writing, and experimental design.

## B Reproducibility and Broader Impact

For reproducibility, we have included code at https://github.com/pujacomputes/datapropsgraphSSL.git. Code is under-development and will be finalized soon. Our code uses the open source torch geometric [53] and PyGCL [54] frameworks.

Self-supervised representation learning is an increasingly popular paradigm for graph representation learning. Critical to many SSL frameworks is the choice of augmentation strategy. As we discuss in this paper, the properties or invariances induced by a particular augmentation strategy are often not well-understood. Failure to understand these properties can lead to unintended effects when the representations are used in downstream tasks. We hope that our work is useful in better understanding the role of augmentations and other data-centric properties on graph representation learning.

## C Extended Discussion

### C.1 Extending our Analysis to other Loss Functions

While our analysis focuses on the spectral contrastive loss (SpecLoss) [15] for ease of exposition, it can also be extended to other contrastive loss functions and predictive methods, such as BYOL [21]. As we noted in Sec. 2, this can be easily accomplished by leveraging our insights on representing graph augmentations through composable graph-edit operations and extending the analyses of Saunshi et al. [41] or Wei et al. [28].

Specifically, the contemporary work of Saunshi et al. proposes a general analysis of contrastive loss functionals and yields a generalization bound similar to Thm. 3.9, e.g., a bound that is dependent on similar data-centric properties and assumptions. In Sec. 3, we decompose GGAs using GED, and then derive expressions for data-centric properties, such as partition dissimilarity, using this decomposition. Since the focus of our analysis is on understanding these data-centric properties in terms of intrinsic dataset attributes (e.g., GED between samples), our theory is complementary to the strategy used by Saunshi et al. Indeed, SpecLoss can be replaced with an alternative contrastive loss functional and adapting the analysis conducted in Sec. 3, we can extend our results to other contrastive losses. For predictive methods, we can leverage recent work by Wei et al. [28] which provides an analysis for unsupervised learning methods for continuous data domains (such as images) by enforcing representation consistency on augmented samples–i.e., BYOL-like methods. Critically, Wei et al.'s generalization analysis relies on properties of the data-generating process's latent space and makes analogous assumptions to the unified recoverability plus separability assumption used in our own work. Thus, our theoretical analysis can be extended to BYOL-like methods by deriving equivalent analytical expressions for the latent-space properties used by Wei et al. Moreover, by representing GGAs using graph edit operations, our derivation of such properties relies upon minimal assumptions and is straight-forward. We do note, however, that Wei et al. assume that the dimension of learned representations is equivalent to the number of classes in the dataset. This can be an invalid assumption in unsupervised learning. In contrast, our analysis is more flexible since we only assume the latent dimension is greater than the number of classes.

## C.2 Evaluation on a Non-Synthetic Dataset

Our analysis in Sec. 3 motivates the need for content-aware augmentations (CAAs) by demonstrating that generic graph augmentations (GGAs) often lead to inconsistent samples, harming representation separability and yielding task *irr*elevant invariances. In Sec. 4.2, we empirically validated these claims in a controlled setting through our new synthetic benchmark and the corresponding oracle CAAs (see Fig. 5). To demonstrate the generality of our analysis in a practical setup, we repeat this experiment in a realistic setting where domain knowledge is available to design content-aware augmentations.

*Experimental Setup.* We analyze BACE, a molecule-protein interaction dataset. We train our models by closely following the setup of Sun et al. [52], who propose biochemistry-inspired augmentations for learning domain-informed representations. In our paper's terminology, these augmentations can be regarded as content-aware augmentations. To ensure fair comparison, we use only "local" CAA, which does not incorporate additional "global" domain knowledge (see Sun et al. [52] for further details). We compare against the strongest GGA baseline reported by the authors, called "mask edge features" augmentation.

For evaluation, we use the trained models to compute the invariance and separability for each sample. As in Sec. 4.2.3, an invariance score is obtained by computing the mean cosine similarity of a sample's representation with 30 of its augmentations. A separability score is computed by dividing the maximum cosine similarity of a given sample and same-class samples by the maximum cosine similarity of a given sample and different-class samples.

*Results.* As demonstrated in Fig. 6, the biochemistry-inspired content-aware augmentations induce much better invariance and separability than the GGA. These results provide further corroboration to our synthetic dataset experiments in 5) and theory in Sec. 3, where we argued that preserving content improves recoverability and leads to task-relevant invariances with better separability.

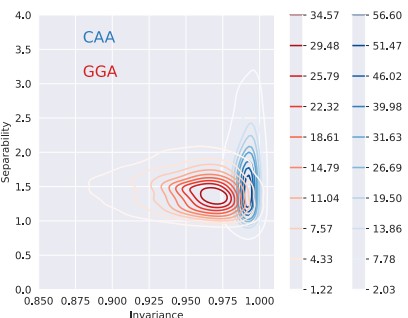

Figure 6: **Invariance vs. Separability**. On BACE [51], a molecule-protein interaction dataset, we compare the content-aware biochemistry-inspired augmentations from MoCL [52] against the GGAs. In this real-world setting, we see that CAAs induce better invariance and separability (Contours are not filled to improve legibility).

## C.3 On Using Mutual Information for Analyzing Task-Relevance in Augmentations

While several different perspectives have been recently proposed for studying self-supervised learning's behavior, many of these frameworks assume that augmentations induce invariance to information that is *irr*elevant to the downstream task, ignoring the potential for augmentations to induce invariance to task-relevant information and harm generalization performance. However, as we discussed in Sec. 2, a notable exception is the information-theoretic analysis of Tian et al. [16]. Specifically, Tian et al. rely upon an information-theoretic framework that interprets the InfoNCE loss as a lower bound of mutual information between two samples. They demonstrate under this framework that optimal augmentations are ones that maximally perturb information irrelevant to the downstream task. However, this viewpoint suffers from the fallacy that InfoNCE is rarely empirically correlated with mutual information. Indeed, Poole et al.[55] demonstrate that this interpretation is only valid when mutual information between two samples is *very* large. For high-dimensional inputs, this will hold true when an augmentation does not alter the input at all, which does not align with the practical behavior of graph (or even image) augmentations. This renders the analysis by Tian et al. relatively inexact compared to our own analysis.

In contrast, we emphasize that our analysis, which has been designed from the ground-up for graph data and augmentations, is more exact. By representing graph augmentations as composable graph-edit distance (GED) operations, we are able to rigorously relate the generalization abilities of a contrastive trained model to intrinsic dataset properties. Specifically, by deriving definitions for partition dissimilarity (Defn 3.8) and inconsistent samples (Lemma 3.6) using GED, our generalization bound relies upon minimal additional assumptions (Thm 3.9). In Sec. 4.2.3 and Sec. C.2, we verify

that our theoretical observations are well supported by our experiments on both synthetic and real-world datasets, further demonstrating the validity of our chosen analysis framework.

# D   Generic Graph Augmentations and Graph Edit Distance

The key insight for our analysis in Sec. 3 is that GGAs can be instantiated in a general manner as a composition of graph edit operations. This allows us to derive a unifying assumption related to recoverability and separability in terms of the graph edit distance (GED) between samples. Here, we provide proofs and additional discussion for the statements made in Sec. 3. We also discuss how our analysis can be interpreted with respect to the population augmentation graph (PAG) proposed by HaoChen et al. [15].

Table 2: Notation

| Symbol | Definition |
|---|---|
| $\overline{\mathcal{X}}$ | The original or natural dataset. |
| $\mathcal{X}$ | Set of all augmented data. |
| $\overline{\boldsymbol{g}} \in \overline{\mathcal{X}}$ | Natural (attributed) graph sample. |
| $\boldsymbol{g}, \boldsymbol{g}' \in \mathcal{X}$ | Augmented (attributed) graph samples |
| $\mathcal{E}_{\overline{\boldsymbol{g}}}$ | Edge set of $\overline{\boldsymbol{g}}$. |
| $\mathcal{V}_{\overline{\boldsymbol{g}}}$ | Node set of $\overline{\boldsymbol{g}}$. |
| $\gamma \in [0, 1]$ | Augmentation strength. Controls the % of edges or nodes that may be perturbed by the selected augmentation. |
| $\mathcal{A}(\overline{\boldsymbol{g}})$ | The set of augmented samples that can be generated from Augmentation, $\mathcal{A}$, given natural sample $\overline{\boldsymbol{g}}$ and $\gamma$. |
| $\mathcal{A}(\cdot \| \overline{\boldsymbol{g}})$ | Distribution of augmentations given a natural sample, $\overline{\boldsymbol{g}}$. |
| $\mathcal{A}(\boldsymbol{g} \| \overline{\boldsymbol{g}})$ | Probability of generating $\boldsymbol{g}$ from $\overline{\boldsymbol{g}}$ given augmentation $\mathcal{A}$. |
| $f$ | Representation Encoder, $f : \{\overline{\mathcal{X}}, \mathcal{X}\} \to \mathbb{R}^d$ |
| $h$ | Classifier, $h : \mathbb{R}^d \to y$ |

## D.1   GGA and Graph Edit Distance

Graph edit distance ($GED$) is used to capture similarity between two graphs. Intuitively, it captures the cost of making elementary edit operations on a graph, $g_1$, to transform it to be isomorphic to another graph, $g_2$. Formally,

**Definition D.1** (Graph Edit Distance (Defn. 3.1)). Let the elementary graph operators (*node insertion*, *node deletion*, *edge deletion*, *edge addition*), and the *categorical feature replacement* operator comprise the set of graph edits. Then, $GED(g_1, g_2) = \min_{(e_1, \dots, e_k) \in \mathcal{P}(g_1, g_2)} \sum_{i=1}^{k} c(e_i)$, where $\mathcal{P}(g_1, g_2)$ is the set of paths (series of edit operations) that transforms $g_1$ to be isomorphic to $g_2$. Here, $e_i$ is $i$-th edit operation in the path, and $c(e_i)$ is the cost for performing the edit.

As shown in Table. 1, elementary graph edit operators can be used to straight-forwardly represent the *node dropping, edge perturbation and subgraph sampling* generic graph augmentations [22]. By introducing an additional graph operator, *categorical feature replacement*, we are also able to consider distance with respect to categorical node attributes. This operator performs a

Table 3: **Generic Graph Augmentations vs. Graph Edit Operators. (Reproduced. Table 1.)** GGA can be straightforwardly expressed using graph edit operators.

| Augmentations | Graph Edit Operators |
|---|---|
| Node Dropping | Node Deletion |
| Edge Perturbation | Edge Deletion, Edge Addition |
| Categorical Attribute Masking | Categorical Feature Replacement Operator |
| Sub-graph Sampling | Node Deletions |

"replacement" whenever there is a disagreement between $g_1$ and $g_2$'s node attributes. Then, the GED is the total cost of structural changes and attribute disagreements between two graphs. Here, we assign a unit cost per operation so all operations are treated equally. Assigning cost to reflect different inductive biases over augmentations is an interesting direction left for future work. Next, we briefly discuss some examples of using graph edit operators to represent GGAs.

Let $(\overline{\boldsymbol{g}}, \boldsymbol{g})$ represent the original and augmented graph respectively, where we perform *node dropping* to obtain $\boldsymbol{g}$. Recall that the *node dropping* augmentation may only drop up to some fraction of nodes in $\overline{\boldsymbol{g}}$. Then, clearly the minimum cost path can then be found using only *node deletion* operators, and the $GED(\overline{\boldsymbol{g}}, \boldsymbol{g})$ is bounded by the number of allowed node drops. Similarly, if $\boldsymbol{g}$ was

obtained through the *edge perturbation* augmentation, which randomly adds or removes a fraction of edges, then $GED(\overline{g}, g)$ is bounded by the number of allowable edge modifications and can be obtained using only *edge addition/deletion* operators. (Here, we allow nodes without edges to still exist, so performing node addition/deletion would not result in a lesser *GED*.) The *sub-graph sampling* augmentation extracts a connected sub-graph that contains at most a fraction of total nodes. The minimum cost path can then be defined using only *node deletions*, e.g. where the operator is applied to all nodes not in the sampled sub-graph. Therefore, $GED(\overline{g}, g)$ is bounded by $|\overline{g}| - |g|$. As discussed above, the *categorical attribute masking* augmentation can be recovered by directly applying the categorical feature replacement operator. Then, the minimum cost path is then the number of differences between the augmented and original samples' node attributes. We formalize the relationships between augmentations and GED in the following Lemmas.

**Lemma D.2.** *Allowable augmentations can be expressed using GED. (Reproduction of Lemma 3.2) Let $\overline{g}$ be a natural sample in $\overline{\mathcal{X}}$, $\mathcal{A}$ be some GGA, $g \sim \mathcal{A}(\cdot|\overline{g})$ be an augmented sample generated from $\overline{g}$ and $\gamma$ be the augmentation strength or the fraction of the graph that GGAs may modify. Then, $\delta \in \{\lfloor\gamma|\mathcal{V}_{\overline{g}}|\rfloor, \lfloor\gamma|\mathcal{E}_{\overline{g}}|\rfloor\}$ represents the number of discrete, allowable modifications for the specified GGA, so $GED(\overline{g}, g) \leq \delta$. Correspondingly, we have $g \in \mathcal{A}(\overline{g}) \Leftrightarrow GED(g, \overline{g}) \leq \delta$.*

*Proof.* Let $\mathcal{P}$ be the shortest path comprised of the edit operators defined in Table. 1 for the given GGA, $\mathcal{A}$. Then, given that at most $\delta$ discrete modifications are permitted and each operator has unit cost, $\text{len}(\mathcal{P}) \leq \delta$ and $\sum_{e_i \in \mathcal{P}} c(e_i) \leq \delta$. Thus, $GED(\overline{g}, g) \leq \delta$. $\square$

**Lemma D.3.** *Upper-bound on Size of Augmentation Set. The size of $A(\overline{g})$ can be upper-bounded through a combinatorial counting process. For example, to determine $A(\overline{g})$ when the considered augmentation is node dropping, we can delineate all sets of possible nodes with size up-to $\gamma|\mathcal{V}_{\overline{g}}|$. Formally, the upper-bound on the number of samples generated using node dropping are:*

$$|\mathcal{A}(\overline{g})| \leq \sum_{j=1}^{\gamma|\mathcal{V}_{\overline{g}}|} \frac{|\mathcal{V}_{\overline{g}}|!}{(|\mathcal{V}_{\overline{g}}| - j)!j!}$$

*We note that this value is an upper-bound because isomorphic pairs are treated as two separate graphs. Furthermore, note the size of the augmentation set grows exponentially with graph size. A similar counting process can be used to determine the number of possible augmented samples obtained through edge perturbation, sub-graph sampling or feature masking. For example, the edge-dropping augmentation could be counted as: $|\mathcal{A}(\overline{g})| \leq \sum_{j=1}^{|\gamma\mathcal{E}_{\overline{g}}|} \frac{|\mathcal{E}_{\overline{g}}|!}{(|\mathcal{E}_{\overline{g}}|-j)!j!}$.*

We further note that because generic graph augmentations (GGAs) perturb the graph randomly, each augmented sample, $g \in \mathcal{A}(\overline{g})$, is equally likely, e.g., $\mathcal{A}(g|\overline{g}) = \frac{1}{|\mathcal{A}|}$.

# E   Details for Generalization Analysis

## E.1   Generalization Analysis

Recently, HaoChen et al. [15] demonstrated that spectral clustering over a graph that captures similarity of augmented data can recover class partitions as augmentations belonging to the same class are more similar, and thus well-connected. These well-aligned partitions can be recovered through spectral decomposition of the similarity graph and the resulting embeddings can be used as features for downstream tasks. The SpecLoss objective, which performs this decomposition, is then defined as follows [15]: Let $g \sim \mathcal{A}(\cdot|\overline{g})$, $g^+ \sim \mathcal{A}(\cdot|\overline{g})$, given $\overline{g} \in \overline{\mathcal{X}}$ and $g^- \sim \mathcal{A}(\cdot|\overline{g}')$, given $\overline{x}' \sim \mathcal{P}_{\overline{\mathcal{X}}} \wedge \overline{g}' \neq \overline{g}$. Then, for the positive/negative pairs $(g, g^+)/(g, g^-)$, the loss $\mathcal{L}(f)$ is:

$$-2 \cdot \mathbb{E}_{g,g^+}\left[f(g)^\top f(g^+)\right] + \mathbb{E}_{g,g^-}\left[\left(f(g)^\top f(g^-)\right)^2\right]$$

By defining SpecLoss through spectral decomposition, its generalization error can be bounded using the recoverability and separability assumptions, which can also be understood in terms of the structure of the similarity graph.

Indeed, in Sec. 3, we demonstrated how GGAs and GED influence recoverability and separability by deriving an analogous generalization bound for SpecLoss that is tailored for graph data. At a

high-level, to find this bound, we derived expressions for recoverability, $\alpha$, and separability, $\rho$, based on graph edit distance, and then used these expression to recover the SpecLoss bound. We then performed some additional manipulation to derive the final expression presented in Thm. 3.9. Here, we provide the details and proofs behind these steps. We begin by restating the Separability plus Recoverability assumption.

**Assumption E.1** (**Separability plus Recoverability Assumption**, (Reproduction of Assm. 3.3)). Let $\overline{g} \in \overline{\mathcal{X}}$ and $y(\overline{g})$ be its label, and $g \sim \mathcal{A}(\cdot|\overline{g})$. Assume that there exists a classifier $h$, such that $h(g) = y(\overline{g})$ with probability at least $1 - \alpha$. We refer to $\alpha$ as the error of $h$.

Now, recall from Sec. 3, that $h$ will incur irreducible error on inconsistent samples, which are defined as follows:

**Corollary E.2.** (*Co-occuring augmentations.,Reproduction of Coll. 3.4*) *Let* $\overline{g} \in \overline{\mathcal{X}}$ *and* $g, g' \in \mathcal{X}$. *Then,* $g \sim \mathcal{A}(\overline{g}) \wedge g' \sim \mathcal{A}(\overline{g}) \Leftrightarrow GED(g, g') \leq 2\delta$, *where* $\delta = \min\{\lfloor \gamma |\mathcal{V}_{\overline{g}}| \rfloor, \lfloor \gamma |\mathcal{E}_{\overline{g}}| \rfloor \lfloor \gamma |\mathcal{V}_g| \rfloor, \lfloor \gamma |\mathcal{E}_g| \rfloor\}$.

*Proof.* Recall, that $g \sim \mathcal{A}(\overline{g}) \iff GED(g, \overline{g}) \leq \delta$ and $g' \sim \mathcal{A}(\overline{g}) \iff GED(g', \overline{g}) \leq \delta$. Then, $GED(g, g') \leq 2\delta$ and are co-occurring augmentations as they both belong to $\mathcal{A}(\overline{g})$. $\square$

**Definition E.3** (**Inconsistent Samples**, Reproduction of Defn. 3.5). Let $g \in \mathcal{X}$, and $y : \overline{\mathcal{X}} \to r$ be a labeling function. Further, let $\overline{\mathcal{X}}_{in} = \{\overline{g}|\overline{g} \in \overline{\mathcal{X}} \wedge GED(g, \overline{g}) \leq \delta\}$ be the set of natural samples that may have generated $g$ and $Y_{in}^* = \{y(\overline{g})|\overline{g} \in \overline{\mathcal{X}}_{in}\}$ be the set of unique labels. If $g$ is an inconsistent sample, $|Y_{in}^*| > 1$.

Now, we fix the behavior of $h$ on inconsistent samples such that $h(g) = y$, for some fixed $y \in Y_{in}^*$. Then, $h$ induces an $r$-way partition over $\mathcal{X}$, such that each sample, $g$, belongs to a partition, $\mathbf{S}_h(g)$. Further, because $h$ will always incur error on inconsistent samples, $\alpha$ can be lower bounded by the ratio of inconsistent to total samples. To this end, we use GED to identify inconsistent samples by identifying disagreement amongst partitions as follows.

**Lemma E.4** (**Using GED to identify inconsistent samples**, Reproduction of Lemma 3.6). *Let* $g, g' \in \mathcal{X}$ *and* $GED(g, g') \leq 2\delta$ *such that* $g \in \mathbf{S}_i \wedge g' \in \mathbf{S}_j$ *and* $i \neq j$, *where partitions are induced by* $h$. *Then, at least one* $\tilde{g} \in \{g, g'\}$ *must be an inconsistent sample.*

*Proof.* By definition, $GED(g, g') \leq 2\delta$ implies that at least one of the following must be true: (i) $\overline{g}_1 \in \overline{\mathcal{X}} \ni y(\overline{g}_1) = i \wedge GED(\overline{g}_1, g) \leq \delta \wedge GED(\overline{g}_1, g') \leq \delta$ or (ii) $\overline{g}_2 \in \overline{\mathcal{X}} \ni y(\overline{g}_2) = j \wedge GED(\overline{g}_2, g) \leq \delta \wedge GED(\overline{g}_2, g') \leq \delta$. WLOG, assume (i). Now, $g' \in \mathbf{S}_j \Leftrightarrow h(g) = j$, so $j \in |Y_{in}^*|$. However, $GED(\overline{g}_1, g) \leq \delta$, so by Lemma 3.2 and Defn. 3.5, $y(\overline{g}_1) = i \in Y_{in}^*$. Since, $i \neq j$, $|Y_{in}^*| > 1$, $g$ must be an inconsistent sample. Note, if (ii) holds, then $g'$ is an inconsistent sample. $\square$

Note that the above lemma does not rely on ground-truth label information to identify inconsistent samples, but only GED from natural samples. Given that the error on inconsistent samples is irreducible, as it is unclear which $y \in Y_{in}$ is correct, we can lower bound the error of $h$ as follows:

**Corollary E.5** (**Error bound due to Inconsistent Samples**, Reproduction of Coll. 3.7). *The error of* $h$ *can be lower-bounded as*

$$\alpha \geq \frac{\sum_i^r \sum_{g \in S_i, g' \notin S_i} \mathbb{1}(GED(g, g') \leq 2\delta)}{|\mathcal{X}|}.$$

Here, the number of inconsistent samples can be approximated via $\sum_i^r \sum_{g \in S_i, g' \notin S_i} \mathbb{1}(GED(g, g') \leq 2\delta)$ and $|\mathcal{X}|$ can be estimated using a combinatorial counting procedure. Thus, the above corollary reflects the fact that error on inconsistent samples cannot be reduced due to label un-identifiability.

Partition dissimilarity, which induces a notion of clustering of similar data-points in our analysis, can be defined as the following:

**Definition E.6** (**Partition Dissimilarity**, Reproduction of Defn. 3.8). Let $S_1, \ldots, S_r$ be an $r$-way partition of $\mathcal{X}$. Then, we define the partition dissimilarity for a given partition as

$$\phi_{\mathcal{X}}(S_i) = \frac{\sum_{g \in S, g' \notin S} \mathbb{1}(GED(g, g') \leq 2\delta)}{\sum_{g \in S} |\{g'|GED(g, g') \leq 2\delta\}|}.$$

We can now state the main result that re-derives the generalization error of SpecLoss in terms of GGAs, using the definitions of co-occurring pairs (Def. 3.4) and dissimilar partitions (Def. 3.8). Notably, we decompose bound in terms of the number of co-occurring augmentation-pairs within the same partition and the number of pairs that cross partitions, which are defined respectively as, $\lambda = \sum_{g \in S_*, g' \in S_*} \math1(GED(g, g') \le 2\delta)$, and $\mu = \sum_{g \in S_*, g' \notin S_*} \math1(GED(g, g') \le 2\delta)$.

**Theorem E.7 (Generalization Bound for SpecLoss with GGA**, Reproduction of Thm 3.9). *Assume the representation dimension $k \ge 2r$ and Assm. 3.7 holds for $\alpha \ge 0$. Let $F$ be a hypothesis class containing a minimizer $f^*_{pop}$ of SpecLoss, $\mathcal{L}(f)$, which produces a $\lfloor k/2 \rfloor$-way partition of $\mathcal{X}$ denoted by $\{S_*\}$. Let its most dissimilar partition have dissimilarity denoted by $\rho_{\lfloor k/2 \rfloor} = \min_i \phi(S_i \in \{S_*\})$. Then, $f^*_{pop}$ has a generalization error bounded as, where the middle term is from the original SpecLoss bound:*

$$\mathcal{E}(f^*_{pop}) \le \widetilde{O}\left(\alpha/\rho^2_{\lfloor k/2 \rfloor}\right) = \widetilde{O}\left(\frac{r}{|\mathcal{X}|}\left[\mu + 2\lambda + \frac{\lambda^2}{\mu}\right]\right),$$

*Proof.* The conversion from recoverability ($\alpha$) and conductance ($\rho$) and within partition ($\mu$) and across partition pairs ($\lambda$), can be derived as follows. We assume that the data distribution is I.I.D and the size of the class partitions are roughly equivalent.

$$\mathcal{E}(f^*_{pop}) \le \widetilde{O}\left(\alpha/\rho^2_{\lfloor k/2 \rfloor}\right) = \widetilde{O}\left(\frac{\sum_i^r \sum_{g \in S_i, g' \notin S_i} \math1(GED(g, g') \le 2\delta)}{|\mathcal{X}|} \frac{1}{\left[\frac{\sum_{g \in S_*, g' \notin S_*} \math1(GED(g,g') \le 2\delta)}{\sum_{x \in S_*} w_x}\right]^2}\right)$$

$$\mathcal{E}(f^*_{pop}) \le \widetilde{O}\left(\alpha/\rho^2_{\lfloor k/2 \rfloor}\right) = \widetilde{O}\left(\frac{\sum_i^r \sum_{g \in S_i, g' \notin S_i} \math1(GED(g, g') \le 2\delta)}{|\mathcal{X}|} \frac{\left[\sum_{x \in S_*} w_x\right]^2}{\left[\sum_{g \in S_*, g' \notin S_*} \math1(GED(g, g') \le 2\delta)\right]^2}\right)$$

$$= \widetilde{O}\left(\frac{r \sum_{g \in S_*, g' \notin S_*} \math1(GED(g, g') \le 2\delta)}{|\mathcal{X}|} \frac{\left[\sum_{x \in S_*} w_x\right]^2}{\left[\sum_{g \in S_*, g' \notin S_*} \math1(GED(g, g') \le 2\delta)\right]^2}\right)$$

$$= \widetilde{O}\left(\frac{r \left[\sum_{x \in S_*} w_x\right]^2}{|\mathcal{X}| \left[\sum_{g \in S_*, g' \notin S_*} \math1(GED(g, g') \le 2\delta)\right]}\right)$$

$$= \widetilde{O}\left(\frac{r \left[\sum_{g \in S_*, g' \notin S_*} \math1(GED(g, g') \le 2\delta) + \sum_{g \in S_*, g' \in S_*} \math1(GED(g, g') \le 2\delta)\right]^2}{|\mathcal{X}| \left[\sum_{g \in S_*, g' \notin S_*} \math1(GED(g, g') \le 2\delta)\right]}\right)$$

$$= \widetilde{O}\left(\frac{r}{|\mathcal{X}|}\left[\frac{\left[\sum_{g \in S_*, g' \notin S_*} \math1(GED(g, g') \le 2\delta)\right]^2}{\left[\sum_{g \in S_*, g' \notin S_*} \math1(GED(g, g') \le 2\delta)\right]}\right.\right.$$

$$+ \frac{2\left[\sum_{g \in S_*, g' \notin S_*} \math1(GED(g, g') \le 2\delta) \sum_{g \in S_*, g' \in S_*} \math1(GED(g, g') \le 2\delta)\right]}{\left[\sum_{g \in S_*, g' \notin S_*} \math1(GED(g, g') \le 2\delta)\right]} + \left.\left.\frac{\sum_{g \in S_*, g' \in S_*} \math1(GED(g, g') \le 2\delta)}{\sum_{g \in S_*, g' \notin S_*} \math1(GED(g, g') \le 2\delta)}\right]\right)$$

$$= \widetilde{O}\left(\frac{r}{|\mathcal{X}|}\left[\sum_{g \in S_*, g' \notin S_*} \math1(GED(g, g') \le 2\delta)\right.\right.$$

$$+ 2 \sum_{g \in S_*, g' \in S_*} \math1(GED(g, g') \le 2\delta) + \left.\left.\frac{\left[\sum_{g \in S_*, g' \in S_*} \math1(GED(g, g') \le 2\delta)\right]^2}{\sum_{g \in S_*, g' \notin S_*} \math1(GED(g, g') \le 2\delta)}\right]\right)$$

$$\tag{4}$$

Now, notice that the above equation can be understood as the number of inconsistent samples vs. the original samples. Let, $\lambda = \sum_{g \in S_*, g' \in S_*} \math1(GED(g, g') \le 2\delta)$ and $\mu = \sum_{g \in S_*, g' \notin S_*} \math1(GED(g, g') \le 2\delta)$. Then, we have recovered the bound presented in Theorem

3.9.

$$\widetilde{O}\left(\alpha/\rho_{\lfloor k/2\rfloor}^2\right) = \widetilde{O}\left(\frac{r}{|\mathcal{X}|}\left[\sum_{\boldsymbol{g}\in S_*,\boldsymbol{g}'\notin S_*}\mathbb{1}(GED(\boldsymbol{g},\boldsymbol{g}')\leq 2\delta)\right.\right.$$

$$\left.\left.+ 2\sum_{\boldsymbol{g}\in S_*,\boldsymbol{g}'\in S_*}\mathbb{1}(GED(\boldsymbol{g},\boldsymbol{g}')\leq 2\delta) + \frac{\left[\sum_{\boldsymbol{g}\in S_*,\boldsymbol{g}'\in S_*}\mathbb{1}(GED(\boldsymbol{g},\boldsymbol{g}')\leq 2\delta)\right]^2}{\sum_{\boldsymbol{g}\in S_*,\boldsymbol{g}'\notin S_*}\mathbb{1}(GED(\boldsymbol{g},\boldsymbol{g}')\leq 2\delta)}\right]\right) \quad (5)$$

$$\approx \widetilde{O}\left(\frac{r}{|\mathcal{X}|}\left[\underbrace{\mu}_{\text{inconsistent samples}} + \underbrace{2\lambda}_{\text{valid samples}} + \frac{\overbrace{\lambda^2}^{\text{valid samples}}}{\underbrace{\mu}_{\text{inconsistent samples}}}\right]\right).$$

Recall, that inconsistent samples can be determined through graph edit distance (Defn. 3.5) between augmented samples. Moreover, that the maximum allowable edit distance between augmented samples is determined by augmentation strength. □

## E.2  Connections to the Population Augmentation Graph

The original bound for SpecLoss uses the population augmentation graph (PAG). While we did not use the PAG in our analysis for ease of exposition, we note that our analysis can be adapted for the PAG as follows:

**Definition E.8** (Population Augmentation Graph [15]). *Let $\mathcal{G}^p$ be the PAG where the vertex set is all augmented data $\mathcal{X}$. For any two augmented data $\boldsymbol{g},\boldsymbol{g}' \in \mathcal{X}$, define the edge weight $w_{\boldsymbol{gg}'}$ as the marginal probability of generating $\boldsymbol{g}$ and $\boldsymbol{g}'$ from a random natural data $\overline{\boldsymbol{g}} \sim \mathcal{P}_{\overline{\mathcal{X}}}$:*

$$w_{\boldsymbol{gg}'} := \mathbb{E}_{\overline{\boldsymbol{g}}\in\mathcal{P}_{\overline{\mathcal{X}}}}[\mathcal{A}(\boldsymbol{g}|\overline{\boldsymbol{g}})\mathcal{A}(\boldsymbol{g}'|\overline{\boldsymbol{g}})]. \quad (6)$$

To extend our analysis to the PAG, we show that connectivity in the PAG is also determined by GED. Then, the definition of inconsistent samples, and partition dissimilarity (conductance) straightforwardly follow.

**Lemma E.9.** *Connectivity in the PAG is determined by GED. Let $\boldsymbol{g},\boldsymbol{g}' \in \mathcal{X}$, and $\overline{\boldsymbol{g}} \in \overline{\mathcal{X}}$. Then, $w_{\boldsymbol{gg}'} > 0 \Leftrightarrow GED(\boldsymbol{g},\boldsymbol{g}') \leq 2\delta$.*

*Proof.* By Lemma 3.4, $w_{\boldsymbol{gg}'} > 0 \Leftrightarrow \mathcal{A}(\boldsymbol{g}|\overline{\boldsymbol{g}}) > 0 \wedge \mathcal{A}(\boldsymbol{g}'|\overline{\boldsymbol{g}}) > 0$. Moreover, if $\mathcal{A}(\boldsymbol{g}|\overline{\boldsymbol{g}}) > 0$ then, $\boldsymbol{g}$ is the augmentation set of $\overline{\boldsymbol{g}}$. If $\boldsymbol{g} \in \mathcal{A}(\overline{\boldsymbol{g}})$ then, $GED(\boldsymbol{g},\overline{\boldsymbol{g}}) \leq \delta$. Then, $w_{\boldsymbol{gg}'} > 0 \Leftrightarrow GED(\boldsymbol{g},\overline{\boldsymbol{g}}) \leq \delta \wedge GED(\boldsymbol{g}',\overline{\boldsymbol{g}}) \leq \delta$, which in turn applies, $w_{\boldsymbol{gg}'} > 0 \Leftrightarrow GED(\boldsymbol{g},\boldsymbol{g}') \leq 2\delta$. □

**Corollary E.10** (**Conductance according to GGA**). *Recall, the conductance $\phi_G$ of a partition $S_i$ in a graph $G$ measures how many edges cross partitions relative to total number of edges a node possesses and that $\mathcal{A}(\boldsymbol{g}|\overline{\boldsymbol{g}}) \approx \frac{1}{|\mathcal{A}(\overline{\boldsymbol{g}})|}$. Then,*

$$\phi_G(S_i) = \frac{\sum_{x\in S, x'\notin S}\mathbb{1}(w_{xx'} > 0)}{\sum_{x\in S} w_x},$$

*where $w_x$ represents the size of $x$'s edge-set.*

Using this definition, we can substitute into the original SpecLoss generalization bound and recover the result presented in Thm. 3.9.

## F  Dataset Generation and Experimental Details

We use the motifs shown in Fig. F to define a 6 class graph classification task. It is important to ensure that the motifs are not isomorphic, as many GNNs are less expressive than the 1-Weisfeiler Lehman's test for isomorphism ([56]). For each class, 1000 random samples are generated as follows: (i) We randomly select between 1-3 motifs to be in each sample. At this time, motifs all belong to the same class, though this condition could easily be changed for a more difficult task. (ii) We define the number of content nodes, $C_n$, as the size of the selected motif, scaled by the

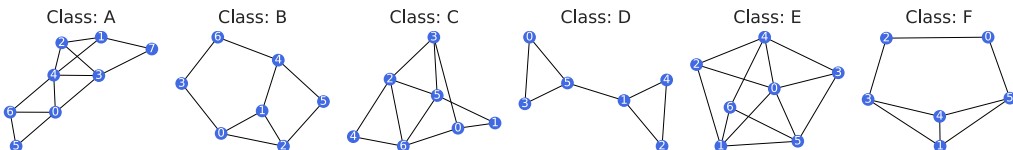

Figure 7: **Motifs used to determine class labels.**

Table 4: *Dataset Description*

| Name | Graphs | Classes | Avg. Nodes | Avg. Edges | Domain |
|------|--------|---------|------------|------------|--------|
| IMDB-BINARY [57] | 1000 | 2 | 19.77 | 96.53 | Social |
| REDDIT-BINARY [57] | 2000 | 2 | 429.63 | 497.75 | Social |
| MUTAG [58] | 188 | 2 | 17.93 | 19.79 | Molecule |
| PROTEINS [59] | 1113 | 2 | 39.06 | 72.82 | Bioinf. |
| DD [60] | 1178 | 2 | 284.32 | 715.66 | Bioinf. |
| NCI1 [61] | 4110 | 2 | 29.87 | 32.30 | Molecule |

number of motifs in the sample. (iii) For a given style ratio, we determine the number of possible style nodes as $S_n = \rho C_n$ (iv). We define $RBG(n)$ using networkx's [2] random tree generator: `networkx.generators.trees.random_tree`. We note that other random graph generators would also be well suited for this task. (v) For additional randomness, we create background graphs using $S_n \pm 2$, and also randomly perturb up-to 10% of edges in sample. We repeat this set-up with $\rho \in \{0.5, 1.0, 1.5, 2.0, 2.5, 3.0, 3.5, 4.0, 4.5, 5.0, 5.5, 6.0, 6.5, 7.5, 8.0\}$ to generate the datasets used in Sec 4.2.

*Experimental Set-up:* We follow You et al. [22] for TUDataset experiments. When reporting the kNN accuracy, we tune $k \in \{5, 10, 15, 20\}$ separately on validation data for each dataset and method to allow for the strongest baselines. For synthetic datasets we use the following setup. Our encoder is a 5-layer GIN model with mean pooling. We set input node features to be a constant 10-dimensional feature vector, and a hidden layer dimension is 32; we concatenate hidden representations for a representation dimension of 160. Models are pretrained for 60 epochs. Subsequently, we use a linear evaluation protocol and train a linear head for 200 epochs. All models are trained with Adam, lr = 0.01.

# G   Related Work

Table 5: **Selected Graph Contrastive Learning Frameworks.** We provide a brief description of augmentations used by selected frameworks. Most frameworks use random corruptive, sampling, or diffusion-based approaches to generate augmentations.

| Method | Augmentations |
|--------|---------------|
| GraphCL ([22]) | Node Dropping, Edge Adding/Dropping, Attribute Masking, Subgraph Extraction |
| GCC ([62]) | RWR Subgraph Extraction of Ego Network |
| MVGRL ([23]) | PPR Diffusion + Sampling |
| GCA ([25]) | Edge Dropping, Attribute Masking (both weighted by centrality) |
| BGRL ([24]) | Edge Dropping, Attribute Masking |
| SelfGNN ([63]) | Attribute Splitting, Attribute Standardization + Scaling, Local Degree Profile, Paste + Local Degree Profile |

*Graph Data Augmentation:* Unlike images, graphs are discrete objects that do not naturally lie in Euclidiean space, making it difficult to define meaningful augmentations. Furthermore, while for images or natural language, there may be an intuitive understanding of what changes will preserve task-relevant information, this is not the case for graphs. Indeed, a single edge change can completely

---

[2]https://networkx.org/documentation/stable/

change the properties of a molecular graph. Therefore, only a few works consider graph data augmentation. [64] note that a node classification task can be perfectly solved if edges only exist between same class samples. They increase homophily by adding edges between nodes that a neural network predicts belong to the same class and breaking edges between nodes of predicted dissimilar classes. However, this approach is expensive and not applicable to graph classification. [30] argue that information preserving topological transformations are difficult for the aforementioned reasons and instead focus on feature augmentations. Throughout training, they add an adversarial perturbation to node features to improve generalization, computing the gradient of the model weights while computing the gradients of the adversarial perturbation to avoid more expensive adversarial training [65]. This approach is not directly applicable to contrastive learning, where label information cannot be used to generate the adversarial perturbation.

*Graph Self-Supervised Learning:*    In graphs, recent works have explored several paradigms for self-supervised learning: see [66] for an up-to-date survey. Graph pre-text tasks are often reminiscent of image in-painting tasks [67], and seek to complete masked graphs and/or node features ([68, 13]). Other successful approaches include predicting auxiliary properties of nodes or entire graphs during pre-training or part of regular training to prevent overfitting ([13]). These tasks often must be carefully selected to avoid negative transfer between tasks. Many contrast-based unsupervised approaches have also been proposed, often inspired by techniques designed for non-graph data. [26, 69] draw inspiration from [9] and maximize the mutual information between global and local representations. MVGRL ([23]) contrasts different views at multiple granularities similar to [8]. [22, 62, 25, 24, 63] use augmentations (which we summarize in Table G) to generate views for contrastive learning. We note that random corruption, sampling or diffusion based approaches used to create generic graph augmentations often do not preserve task-relevant information or introduce meaningful invariances.