# OpenReview forum: "Analyzing Data-Centric Properties for Graph Contrastive Learning"
_NeurIPS.cc/2022/Conference — NeurIPS 2022 Accept_

### Official Review · Reviewer_SttW · 2022-07-04

**Rating:** 5
**Confidence:** 4
**Soundness:** 3 good
**Presentation:** 2 fair
**Contribution:** 3 good

**Summary:**

This paper theorectically demonstrates why task-relevant graph augmentation is required, using the concept of Graph Edit Distance (GED). In experiments, they showed that existing generic graph augmentation induce limited task-relevent invariance, revealing the error of common graph augmentation practice. To prove their theoretical analysis, they generate synthetic data and show content aware augmentation (task relevant augmentation) improves model performance.

**Questions:**

- In Assumption 3.3, can the similarity of representation be guaranteed simply with correct classification?
- Is this theoretical analysis only applicable for SpecLoss?

**Limitations:**

This work provides theoretical analysis in common practice of graph augmentation. However, it would be better if the authors could validate the practicality of the proposed method, i.e., to real-world graphs.

**Strengths And Weaknesses:**

Strengths
- This paper has strong theoretical background.
- Also, to my best knowledge, this work is the first attempt to analyze graph augmentation in terms of Graph Edit Distance (GED).
- Authors demonstrate their theoretical analysis through experiments on synthetic datasets.

Weakness
- Paper is hard to read. Specifically, concept of invariance, recoverability and separability should be more clearly defined. Moreover, definition of task relevant augmentation / augmentation that induces invariance to task-irrelevant attributes is not clear.
- Practicality is limited. Different from image data, it is hard to dichotomously divide style and content in graph domain. I think this is the reason why authors couldn't show the correlation between invariance and accuracy in real-world dataset.

---

> ### Author Response · Authors · 2022-08-02
> **Response to Reviewer SttW (Part 1)**
>
> Thank you for the helpful feedback! We are pleased to see that our work was found to have “strong theoretical background” and recognized as “the first attempt” to theoretically analyze graph augmentation. Below, we address all questions and concerns.
>
> ***Response to Limitations:***
>
> > However, it would be better if the authors could validate the practicality of the proposed method, i.e., to real-world graphs.
>
> Thank you for this comment! Following reviewer suggestion, ***we now include new results (Fig. 6, Supp. B3) on BACE, a molecule-protein interaction dataset, to demonstrate the practicality of our analysis.*** We compare generic graph augmentations against the biochemistry-based augmentations proposed by [1], which can be regarded as content-aware augmentations in this scenario. ***The results and conclusions on this real-world graphs dataset strongly corroborate with those from our synthetic dataset experiments*** (Fig. 4) -- use of content-aware augmentations improves recoverability, leads to task-relevant invariances with better separability, and yields better application performance.
>
>
> [1] MoCL: Contrastive Learning on Molecular Graphs with Multi-level Domain Knowledge, KDD 2021.
>
> ***Response to Questions:***
>
> > In Assumption 3.3, can the similarity of representation be guaranteed simply with correct classification?
>
> Assumption 3.3 formalizes which representations should be expected to be similar and quantifies how similar they are on average using $\alpha$, the classification error of a linear probe. We use “correct classification” as a measure for similarity of representations in the assumption simply because it makes for an easier theoretical analysis. One can certainly define alternative, conceptually equivalent versions of this assumption (may be more difficult to analyze).
>
> > Is this theoretical analysis only applicable for SpecLoss?
>
> Thank you for the question! Our analysis can indeed be extended to other methods. In brief, we first note one of our main contributions includes recasting graph augmentations into composable graph edit-distance (GED) operations. This allows us to controllably estimate the effects of recoverability and separability on the generalizability of a model trained with contrastive learning using SpecLoss. As we allude to at L117, contemporary work [1] develops a general analysis of contrastive loss functionals and yields a generalization bound similar to ours. Since our focus is on the data-centric properties, our theory is complementary to the strategy used by Saunshi et al. in [1]. ***Hence, by replacing SpecLoss with a contrastive loss functional and following the exact same analysis as we devised in our paper, we can extend our results to any contrastive loss!*** Similarly, for methods like BYOL, we highlight the recent paper by Wei et al. [2], who provided an analysis for unsupervised learning in continuous data domains (such as images) by enforcing representation consistency on augmented samples. Their theory relies on properties of the data-generating process's latent space and makes recoverability and separability assumptions, similar to ours. Thus, to extend our analysis to BYOL-like methods, one need only derive analytical expressions for the latent-space properties used by Wei et al. [2]! This derivation is relatively straightforward, but we do stress that the analysis of Wei et al. assumes the learned representations have a dimension equal to the number of classes in the dataset. This can be an invalid assumption in unsupervised learning. In contrast, our analysis is more flexible since we only assume the latent dimension is greater than the number of classes, which is generally true for most practical scenarios.
>
> [1] Nikunj Saunshi, Jordan Ash, Surbhi Goel, Dipendra Misra, Cyril Zhang, Sanjeev Arora, Sham Kakade, and Akshay Krishnamurthy. Understanding contrastive learning requires incorporating inductive biases, In ICML, 2022.
>
> [2] Colin Wei, Kendrick Shen, Yining Chen, and Tengyu Ma. Theoretical analysis of self-training with deep networks on unlabeled data. In ICLR, 2021.

---

> > ### Author Response · Authors · 2022-08-02
> > **Response to Reviewer SttW (Part 2)**
> >
> > ***Response to Weaknesses***
> > > Paper is hard to read. Specifically, the concept of invariance, recoverability and separability should be more clearly defined. Moreover, definition of task relevant augmentation / augmentation that induces invariance to task-irrelevant attributes is not clear.
> >
> > Thank you for the suggestion! *To aid better understanding, we have now added Fig. 5 in Supplementary B1.* We will ensure this figure is shifted to the main paper in the final version. Here, we briefly recapitulate the extended discussion:
> >
> > - *Separability:* This property states that there is an underlying separation between the manifolds instantiating datapoints of different classes. Formally, we instantiate this property by assuming there exists a latent space where the optimal classifier $h$ is able to classify “natural” (i.e., unaugmented) samples with a small error.
> >
> > - *Recoverability:* This property states that augmentations of a sample remain close to its related samples from the same class on the underlying data manifold. This ensures that via augmentations, one does not convert samples from one class to another. Formally, we instantiate this property by assuming the classifier $h$ is able to assign an augmented sample to the same class as that of its natural sample.
> > - *Invariance:* Arguably, given the right training objective and sufficient model capacity, one can retrieve a representation that is invariant to any arbitrary transformation. Formally, we instantiate this invariance by assuming the global minimizer of a contrastive loss outputs the same representation for a natural sample and its augmented version.
> >
> > - *Task-relevance:*  Not all invariances are useful for a given task. For ex., when classifying different labrador breeds, color invariance makes it hard to discriminate between chocolate vs. yellow labradors. Our analysis implicitly defines task-relevant augmentations via the frequency of “inconsistent” samples, e.g., augmentations that are likely to be generated from two different classes. Such samples are confusing to classify for even optimal classifiers. Our analysis formally demonstrates how inconsistent samples harm generalization error.

---

> > > ### Author Response · Authors · 2022-08-02
> > > **Response to Reviewer SttW (Part 3/Summary)**
> > >
> > > > Practicality is limited. Different from image data, it is hard to dichotomously divide style and content in the graph domain. I think this is the reason why authors couldn't show the correlation between invariance and accuracy in real-world dataset.
> > >
> > > We note that our focus was on understanding graph CL and its interplay with graph augmentations, which are notoriously hard to model. ***Our theoretical contribution of recasting graph augmentations via composable graph-edit distance operators already provides insights into how generic graph augmentations lead to poor representation quality and opens the avenue for future theoretical research on graph CL.***
> > >
> > > Moreover, our theoretically-grounded synthetic dataset and analysis hold practical value. For example, consider automated augmentation methods based on bi-level or adversarial optimization, which seek to automatically find augmentations for unsupervised graph representation learning (e.g., JOAO and AD-GCL). Due to the lack of a ground-truth set of useful augmentations for a dataset, it has been difficult to ascertain if such techniques actually work: i.e., do they identify an optimal set of augmentations?” ***Through our thoughtfully designed synthetic dataset, we show automated augmentations methods often fail to find task-relevant augmentations and hence generalize poorly as the style ratio changes (Fig. 3).*** To our knowledge, these results are the first grounded insights into automated augmentation techniques.
> > >
> > > Finally, we agree with the reviewer that it can be difficult to intuitively divide style vs. content for real-world graph applications. However, we strongly emphasize that there is generally enough domain information available to design useful content-aware augmentations. ***Indeed, in an additional experiment (see Fig 6, Supp. B3), we use the content-aware augmentations proposed by MoCL [1] to show that preserving content does lead to useful, task-relevant invariances in a realistic molecular representation learning use-case.*** We also stress that, while our intuitive understanding of natural images can make it appear that such specialized knowledge is not necessary for vision tasks, this is not the case at all! In fact, [2] demonstrates that naively training with standardly used image augmentations induces occlusion invariance, which severely hurts performance if the downstream task is object detection, where viewpoint invariance is more important.
> > >
> > > *Summary:* Fig. 2 demonstrates that accuracy and invariance are indeed uncorrelated, and that generic graph augmentations are unable to disentangle style and content. However, when domain knowledge is used to design content-aware augmentations, as in MoCL [1] or using oracle augmentations on our synthetic dataset, we see that both invariance and accuracy are improved. These results highlight that the style vs. content perspective is valuable for studying graph augmentations, and supports the practicality of our approach.
> > >
> > > [1] MoCL: Contrastive Learning on Molecular Graphs with Multi-level Domain Knowledge, KDD 2021.
> > >
> > > [2] Senthil Purushwalkam and Abhinav Gupta. Demystifying contrastive self-supervised learning: Invariances, augmentations and dataset biases, In NeurIPS, 2020
> > >
> > > --------
> > > **Overall Summary:** Thank you for your feedback and comments which have helped us better present our contributions! As we discussed above, our work rigorously considers the role of data-dependent properties, e.g., invariance, recoverability and separability, in the generalization behavior of graph contrastive learning. Indeed, our novel generalization analysis not only provides the first formal framework for studying graph augmentations in graph contrastive learning, it is also empirically supported by experiments on both synthetic and real-world datasets, and easily extended to other contrastive or predictive methods. Furthermore, our principled synthetic data generator provides novel insights into advanced augmentation methods and will be a useful benchmark for the graph SSL community. In case our answers have justifiably addressed your concerns, we respectfully request that you increase your score to support the acceptance of our work.

---

> ### Author Response · Authors · 2022-08-06
> **Gentle Reminder**
>
> Dear Reviewer SttW,
>
> We hope that we have justifiably answered your questions and properly accommodated the feedback in your initial review. Since the discussion period ends soon, please let us know if you have any further questions. Thank you!

---

> > ### Comment · Reviewer_SttW · 2022-08-09
> > **Thanks for the authors’ kind response to our concerns.**
> >
> > Thanks for the authors’ kind response to our concerns.
> >
> > Confusing terms (such as recoverability and separability ) seem to be well organized through Figure 5.
> > And extensibility of theorectical analysis in various loss term seems interesting.
> >
> > However, still my concern on practicality has not been well addressed.
> > Although MoCL is good example of content-aware augmentation, it still requires domain knowledge which is hard to find in real world.
> >
> > However, I still think attempting to analyze graph augmentation in terms of GED is novel. I’ll keep positive side on this paper.

---

> > > ### Author Response · Authors · 2022-08-09
> > > **Thank you, and Discussion of Praticality**
> > >
> > > Thanks you for your positive assessment of our work! We are glad that our rebuttals and addition of Fig. 5 effectively clarified your concerns, and you found extensions of our framework to other loss functions interesting. We address the remaining question about practicality below.
> > >
> > > **Practicality**: We respectfully stress that ***demonstrating the limitations and pitfalls of a technique is equally valuable to practitioners as proposing a new method***. Indeed, the practicality of our analysis lies in the fact that it formally demonstrates the need for including domain-specific augmentation priors in graph self-supervised learning. Further, in a spirit similar to your argument that domain specific knowledge may sometimes be difficult to access in some scenarios, prior works have proposed automated augmentation techniques. However, our work shows that such ***"heuristically" motivated automated-augmentations techniques are not a viable augmentation strategy***: as we thoroughly illustrate via our novel synthetic benchmark, automated augmentations fail to find useful invariances! This results further motivates the need for domain-aware augmentation strategies. Moreover, we note this ***proposed benchmark itself can have immense practical value*** for future research in graph self-supervised learning--e.g., by serving as a mainstay evaluation setting for alternative augmentations paradigms!

---

### Official Review · Reviewer_1y4x · 2022-07-12

**Rating:** 7
**Confidence:** 3
**Soundness:** 4 excellent
**Presentation:** 3 good
**Contribution:** 3 good

**Summary:**

The paper performs systematic and quantitative analyses on the relationship between properties of generic graph augmentations and model performance by deriving the generalization error bound. Analyses and empirical results show that commonly used GGAs do not necessarily lead to task-relevant invariance. Further experimental studies with proposed synthetic data generation indicate benefits from recoverable augmentations.

**Questions:**

The analysis specifically focuses on the SpecLoss. How would specifically implemented labeling functions and learning objectives (quantitatively) affect the error bound (or can their error bound be derived specifically)? For example, BYOL is different to typical contrastive methods as it does not require negative pairs but experimental studies still include BYOL.

**Strengths And Weaknesses:**

\+ To my knowledge, while adopting existing concepts such as recoverability and separability as a tool, the paper performs the first study to quantitatively study the relationship between measured augmentation properties and the performance (error bound) of SSL methods.

\+ The proposed theory and its derivation is quite solid. And the empirical results are insightful.

\- The authors claim that the analysis motivate the necessity of ask-aware augmentations one of their contributions. In fact, the conclusion has been derived from existing studies such as Tian et al. [16] from the mutual information prospect of view, i.e., the worst performance is achieved when there is no task-relevant mutual information overlapping MI of two views. A discussion on the differences and connection of the two groundings could strengthen this as a contribution.

\- The study mainly focus on the theoretical analysis specifically regarding the GGA in contrastive learning (SpecLoss). The title can be made more specific accordingly to avoid potential over-claim or misleading. Moreover, while strict derivation is not necessary, it'd be better to include discussions regarding how/whether the recoverability and separability can be adopted in predictive methods (or objectives other than SpecLoss) with augmentations (e.g., Section 4.2 of [1*]).

[1*] Xie, Yaochen, et al. "Self-supervised learning of graph neural networks: A unified review." IEEE Transactions on Pattern Analysis and Machine Intelligence (2022).

---

> ### Author Response · Authors · 2022-08-02
> **Response to Reviewer 1y4x (Part 1)**
>
> Thank you for your insightful feedback! We are delighted to see that the novelty of our framework is appreciated as “the first study to quantitatively study” the relationship between  augmentations and the generalization error of SSL methods. We are further encouraged that our results were found “insightful” and “solid.” Below, we address all questions and concerns.
>
> > The authors claim that the analysis motivates the necessity of task-aware augmentations as one of their contributions. In fact, the conclusion has been derived from existing studies such as Tian et al. [16] from the mutual information prospect of view, i.e., the worst performance is achieved when there is no task-relevant mutual information overlapping MI of two views. A discussion on the differences and connection of the two groundings could strengthen this as a contribution.
>
> Thank you for the suggestion! We begin by highlighting that our paper is the first to represent graph augmentations as composable graph-edit distance (GED) operations, rigorously relating generalization abilities of a contrastively trained model to intrinsic graph dataset properties. This allows us to formally understand the limitations of generic graph augmentations (GGAs), as well as motivate why task-aware graph augmentations are necessary. ***We strongly emphasize that our analysis is precise since we derive exact definitions of GGAs using GED, and correspondingly our generalization bounds rely on minimal assumptions (Thm 3.9).***
>
> In contrast, as noted by the reviewer, the analysis performed by Tian et al. [1] relies on an information-theoretic framework that is based on the interpretation of the InfoNCE loss as a lower bound of mutual information between two samples. However, this interpretation is only valid when mutual information between two samples is very large (see Poole et al. [2]). For high-dimensional inputs, this essentially implies that an augmentation does not alter the input at all! *This renders the analysis by Tian et al. relatively inexact compared to our own analysis,* since their assumptions are incorrect in practice. We note we have also added the above discussion to the supplementary (Supp. B4).
>
> [1] Yonglong Tian, Chen Sun, Ben Poole, Dilip Krishnan, Cordelia Schmid, Phillip Isola. What Makes for Good Views for Contrastive Learning?, In NeurIPS, 2021.
>
> [2] Ben Poole, Sherjil Ozair, Aaron van den Oord, Alexander A. Alemi, George Tucker. On Variational Bounds of Mutual Information, In ICML, 2019.
>
>
> > The study mainly focuses on the theoretical analysis specifically regarding the GGA in contrastive learning (SpecLoss). The title can be made more specific accordingly to avoid potential over-claim or misleading.
>
> Thank you for the suggestion! We’ve updated the title and abstract to precisely indicate that our theory focuses on contrastive learning.

---

> > ### Author Response · Authors · 2022-08-02
> > **Response to Reviewer 1y4x (Part 2)**
> >
> > > Moreover, while strict derivation is not necessary, it'd be better to include discussions regarding how/whether the recoverability and separability can be adopted in predictive methods (or objectives other than SpecLoss) with augmentations (e.g., Section 4.2 of [1]).
> >
> > Thank you for the comment! We've added a discussion in section Supp. B2 on how our analysis can be extended to other loss functions and predictive methods. To summarize, we first note one of key insights is the recasting graph augmentations into composable graph edit-distance (GED) operations. This allows us to controllably estimate the effects of recoverability and separability on the generalizability of a model trained with contrastive learning using SpecLoss. As we allude to at L117, contemporary work [1] develops a general analysis of contrastive loss functionals and yields a generalization bound similar to ours. Since our focus is on the data-centric properties, our theory is complementary to the strategy used by Saunshi et al. in [1]. ***Hence, by replacing SpecLoss with a contrastive loss functional and following the exact same analysis as we devised in our paper, we can extend our results to any contrastive loss!***
> >
> > Similarly, for methods like BYOL, we highlight the recent paper by Wei et al. [2], who provided an analysis for unsupervised learning methods for continuous data domains (such as images) by enforcing representation consistency on augmented samples--i.e., using methods like BYOL. Their theory relies on properties of the data-generating process's latent space and makes assumptions essentially the same as our own recoverability and separability assumptions. ***Thus, to extend our theoretical analysis to BYOL-like methods, one need only derive analytical expressions for the latent-space properties used by Wei et al. [2]! This derivation is a relatively straightforward extension of our current analysis***, but we do stress that the analysis of Wei et al. assumes the learned representations have a dimension equal to the number of classes in the dataset. This can be an invalid assumption in unsupervised learning. In contrast, our analysis is more flexible since we only assume the latent dimension is greater than the number of classes, which is generally true for most practical scenarios.
> >
> > *Summary:* Our analysis can be easily extended to other loss functions and predictive methods, such as BYOL, by leveraging our insight on representing graph augmentations as composable graph-edit operations and using the analysis of Saunshi et al. [1] or Wei et al. [2], who respectively derive generalization bounds for general contrastive loss functions and BYOL-like methods.
> >
> > [1] Nikunj Saunshi, Jordan Ash, Surbhi Goel, Dipendra Misra, Cyril Zhang, Sanjeev Arora, Sham Kakade, and Akshay Krishnamurthy. Understanding contrastive learning requires incorporating inductive biases, In ICML, 2022.
> >
> > [2] Colin Wei, Kendrick Shen, Yining Chen, and Tengyu Ma. Theoretical analysis of self-training with deep networks on unlabeled data. In ICLR, 2021.
> >
> >
> > > How would specifically implemented labeling functions and learning objectives (quantitatively) affect the error bound (or can their error bound be derived specifically)? For example, BYOL is different to typical contrastive methods as it does not require negative pairs but experimental studies still include BYOL.
> >
> > Please see our answer above! In brief, we can exploit the analysis by Wei et al. [1] and, via our analysis of graph augmentations as composable graph-edit operations, derive analytical expressions for their assumed properties/constraints on the data-generating process. This is straightforward since the properties discussed by Wei et al. [1] are related to the separability plus recoverability assumption (Assm. 3.3) used in our paper.
> >
> > [1] Colin Wei, Kendrick Shen, Yining Chen, and Tengyu Ma. Theoretical analysis of self-training with deep networks on unlabeled data. In ICLR, 2021.

---

> ### Author Response · Authors · 2022-08-06
> **Gentle Reminder**
>
> Dear Reviewer 1y4x,
>
> We hope that we have justifiably answered your questions and properly accommodated the feedback in your initial review. Since the discussion period ends soon, please let us know if you have any further questions. Thank you!

---

### Official Review · Reviewer_kxe9 · 2022-07-19

**Rating:** 6
**Confidence:** 3
**Soundness:** 3 good
**Presentation:** 4 excellent
**Contribution:** 3 good

**Summary:**

This paper investigates how can graph SSL methods work well. Specifically, they first perform a generalization analysis for generic graph data augmentations based on dataset recoverability and separability constraints. Also, they observe that generic graph augmentations adopted in GraphCL do not introduce meaningful invariance. And then, they introduce a designed data generation process to control the amount of content vs. style in samples, which enables to empirically evaluate oracle augmentations.

**Questions:**

See the weakness above.

**Limitations:**

The authors has adequately addressed the limitations and potential negative societal impact of their work.

**Strengths And Weaknesses:**

**Strengths**

[+] The paper is well-organized and easy to follow.

[+] The empirical and analytical framework is reasonable.

[+] The codes for reproducing the reported results are provided.

[+] This work shows some insightful opinions on data augmentations in graph contrastive learning. Also, they provide a useful benchmark to better evaluate oracle augmentations.

**Weakness**

[-] Previous works on content-aware augmentations [1] or implicit augmentations [2] in graph contrastive learning should be discussed.

[-] The technical contributions are limited. The content-aware augmentations in this paper only work well in the generated toy dataset.  How can the empirical and analytical framework help us develop more effective graph data augmentations or more powerful graph URL algorithms for practical graph data?

[-] Studying data augmentations in self-supervised learning from a 'content vs. style' perspective has been performed in previous works [3].

[-] I notice that the codes (Readme file) uploaded by the authors include some personal information. If the authors use the open-sourced pyGCL framework, they should cite the paper [4] to violate the anonymity rules.

[1] MoCL: Contrastive Learning on Molecular Graphs with Multi-level Domain Knowledge, KDD 2021.

[2] SimGRACE: A Simple Framework for Graph Contrastive Learning without Data Augmentation, WWW 2022.

[3] Self-Supervised Learning with Data Augmentations Provably Isolates Content from Style, NeurIPS 2021.

[4] An Empirical Study of Graph Contrastive Learning, NeurIPS 2021 (Datasets and Benchmarks track)

---

> ### Author Response · Authors · 2022-08-02
> **Response to Reviewer kxe9 (Part 1)**
>
> We thank the reviewer for their detailed comments! We are encouraged to see that they found our empirical and analytical framework “reasonable”, our observations “insightful” and our proposed benchmark “useful.” Below, we address all questions and concerns.
>
> > Previous works on content-aware augmentations [1] or implicit augmentations [2] in graph contrastive learning should be discussed.
>
> Thank you for the suggestion! We have updated our paper and supplementary to include discussions of [1] and [2] and summarize our changes here.
>
> - **Discussion on [1]**: We now include a discussion in Sec. 4.2.3 and Supp. B3  on MoCL [1], which proposes the use of biochemistry knowledge to define content-aware augmentations for representation learning on molecules. More importantly, we now include ***further results*** illustrating the use of content-aware augmentations in MoCL leads to representations that have better invariance and separability than representations learnt using generic graph augmentations (see Fig. 6, Supp. B3). These results ***directly support*** our synthetic dataset experiments in Fig. 4 and theory in Sec. 3, where we demonstrate preserving content improves recoverability and leads to task-relevant invariances with better separability. ***Beyond providing further corroboration to our theoretical and empirical findings in a realistic setting, this experiment also highlights the practicality of our analysis.***
>
> - **Discussion on [2]**: We now include a citation in related work (Sec. 2) for SimGRACE [2], which proposes the perturbation of model weights as a form of "implicit" augmentation. More importantly, ***we have added experimental results on SimGRACE (Fig. 3) to illustrate our analysis generalizes to yet another paradigm of augmentations.*** In brief, despite tuning the perturbation magnitude and learning rate, we find that the implicit augmentation strategy proposed in SimGRACE is not able to induce style-invariance (evidenced by the decrease in performance with increase in the style-content ratio). Additionally, we see that SimGRACE's performance trails behind GraphCL with generic graph augmentations (20/%) and AD-GCL, suggesting that in absence of content-aware augmentations, other methods may be better suited. We also note that while perturbing the model weights does avoid corrupting the input space, it is not clear what invariances are learnt by the model. This can decrease the interpretability of the model, and, as shown in Fig. 3, can decrease generalizability if the implicit augmentations are not able to recover style invariance in the input-space.
>
> *Summary:* Following reviewer suggestions, we have added discussions and additional experiments on MoCL [1] and SimGRACE [2] to the paper. Our experiments further validate our analysis in a realistic setting (representation learning on molecules) and help extend our results to yet another class of augmentations ("implicit" or weight perturbation augmentations). Both these results illustrate the utility and the generality of our analysis.

---

> > ### Author Response · Authors · 2022-08-02
> > **Response to Reviewer kxe9 (Part 2)**
> >
> > >The technical contributions are limited. The content-aware augmentations in this paper only work well in the generated toy dataset. How can the empirical and analytical framework help us develop more effective graph data augmentations or more powerful graph URL algorithms for practical graph data?
> >
> > Thank you for your feedback! To highlight our technical contributions and the utility of our framework, we describe below how our work contributes the ***first generalization analysis of CL*** when using popular generic graph augmentations (GGAs), and provides ***valuable empirical insights using our thoughtfully designed synthetic benchmark.*** We firmly believe that these frameworks will be useful to the community in evaluating and understanding novel graph URL algorithms and augmentations.
> >
> > - **Novel Theoretical Analysis.** To the best of our knowledge, our theoretical analysis is the first to provide a framework for studying the generalization of contrastive learning when using GGAs. Moreover, by decomposing GGAs using graph edit operators, our analysis relies upon minimal assumptions and is dependent only upon the graph edit distance between samples, a property intrinsic to the dataset--i.e., in contrast to prior analyses of contrastive learning, our analysis is exact because it was intentionally designed from the ground-up to work with graph data. Finally, we strongly emphasize that ***our framework can provide mathematical justification for the intuition and heuristics*** already known to the community. For example, we show that the phenomenon of optimal augmentation strength can be understood through changes in recoverability and separability in Sec. 3.
> >
> > - **Evaluation of Automated/Advanced Augmentation Methods.** Recently, automated augmentation methods based on bi-level or adversarial optimization have been proposed to automatically find augmentations for graph URL (e.g., JOAO and AD-GCL). Due to the lack of a ground-truth set of useful augmentations for a dataset, it has been difficult to ascertain if such techniques identify the optimal augmentations that will yield the most powerful representations that generalize across varying “styles.” ***By providing a controlled setting via our proposed synthetic dataset, we show for the first time that in fact automated augmentations methods often fail to find task-relevant augmentations.*** As a result, we find in Fig. 3 that automated methods fail to generalize as the style ratio changes, since they cannot identify task-relevant augmentations! In fact, using our proposed benchmark we found that even when ground-truth, content-aware augmentations are included in JOAO's augmentation pool, JOAO prefers to select task-irrelevant augmentations that lead to poor generalization! To our knowledge, these results are first-ever grounded insights into automated graph augmentation techniques; we strongly believe our benchmark will provide a useful, controlled setting for evaluation of future, automated augmentation methods by the community. For example, by taking the reviewer's suggestion into account, we have been able to provide a detailed account of limitations of yet another augmentation paradigm: SimGRACE and implicit augmentations. Specifically, as we discussed above, we find that SimGRACE does not induce task-relevant invariance and witnesses limited performance as style changes.
> >
> > *Summary:*
> > Our proposed theoretical and empirical framework makes several technical contributions which we firmly believe will be useful to the community in understanding the behavior of both common and advanced augmentation practices. We strongly emphasize that our theoretical analysis is exact, tailored for graphs, and the first of its kind for understanding contrastive learning with graph augmentations. Moreover, we highlight that our synthetic benchmark provides a useful benchmark for evaluating advanced augmentation techniques, via which we were able to demonstrate for the first time simple failure cases in automated augmentation solutions that were not apparent when using standard benchmarks.
> >
> > > Studying data augmentations in self-supervised learning from a 'content vs. style' perspective has been performed in previous works [3].
> >
> > Please note we already highlight in the paper that our synthetic data generation process was *partially* motivated by the analysis conducted by [3]. However, we strongly emphasize that [3]’s focus was to theoretically argue that content vs. style disentanglement is emergent in self-supervised learning methods. In contrast, our focus is to develop a grounded benchmark that provides adjustable knobs over content (task-relevant) and style (task-irrelevant) information, allowing us to evaluate how different graph SSL algorithms benefit from content-aware augmentations and compare them against oracle augmentations.

---

> > > ### Author Response · Authors · 2022-08-02
> > > **Response to Reviewer kxe9 (Part 3/Summary)**
> > >
> > > > I notice that the codes (Readme file) uploaded by the authors include some personal information. If the authors use the open-sourced pyGCL framework, they should cite the paper [4] to violate the anonymity rules.
> > >
> > > Thanks for the catch! We forked pyGCL from Github and left the original ReadME by mistake. We strongly emphasize we are **NOT** the authors of pyGCL, and have updated the ReadME/main paper accordingly.
> > >
> > > --------
> > >
> > > **Overall Summary:**  Thank you for your comments and suggestions that have helped us better contextualize our contributions! As we discussed above, our work rigorously considers the role of data-dependent properties, e.g., invariance, recoverability and separability, in the generalization behavior of graph contrastive learning. Indeed, we provide the first generalization analysis of graph contrastive learning with popularly used GGAs, and validate our observations using the real-world content-aware augmentations proposed by MoCL. Moreover, we propose a synthetic benchmark that will be useful to the graph SSL community. In case our answers have justifiably addressed your concerns, we respectfully request that you increase your score to support the acceptance of our work.

---

> ### Author Response · Authors · 2022-08-06
> **Gentle Reminder**
>
> Dear Reviewer kxe9,
>
> We hope that we have justifiably answered your questions and properly accommodated the feedback in your initial review. Since the discussion period ends soon, please let us know if you have any further questions. Thank you!

---

> > ### Comment · Reviewer_kxe9 · 2022-08-08
> > **Thanks for the authors' response**
> >
> > Thanks for the authors' detailed response, which addresses my concerns a lot!  Hence, I raise my rating from 5 to 6 (weak accept).

---

### Author Response · Authors · 2022-08-02
**Contributions and Summary of Paper Updates**

Thank you to all the reviewers for their positive and insightful feedback! In the following, we summarize our contributions and report the steps taken during the rebuttals phase to revise the paper:

- **Contributions:** We re-emphasize that our paper is the first to provide a framework for studying the generalization of contrastive learning when using popular generic graph augmentations. Our analysis, in fact, motivates a synthetic data generation process that serves as a useful benchmark for evaluating different methods and augmentation paradigms. Indeed, via this dataset, we were able to show several limitations in the abilities of advanced augmentation methods that use bilevel optimization or implicit techniques such as weight perturbations.

- **Additional experiments:** Based upon reviewer kxe9's suggestions, we have added additional discussion around recommended papers on content-aware and implicit augmentations. More importantly, we have added new experiments to incorporate the techniques proposed in these papers with our results (see Fig. 6 and Fig. 3). Given the continued success of our claims with these addition experiments, we clearly see further corroboration of our analysis in real-world settings (requested by reviewer sttw) and further verify usefulness of our synthetic dataset as a test-bed for identifying limitations in advanced augmentation paradigms.

- **Further discussion:** We included additional discussion on: (i) the straight-forward extension of our analysis to other loss functions, (ii) the limitations of other theoretical SSL frameworks (Tian et al.’s), and (iii) definitions of data-centric properties need for our analysis (see Fig. 5).

---

### Meta-Review · Area_Chair_14aP · 2022-08-23

**Recommendation:** Accept
**Confidence:** Certain

**Metareview:**

Overall the reviews are positive, appreciating the proposed theory, derivation, and presentation. Also some concerns raised are properly addressed during the discussion period. Hence, I recommend the accetance of this paper.

**Award:**

No

---

### Decision · Program_Chairs · 2022-09-14

Accept